# Integrative analysis reveals RNA G-quadruplexes in UTRs are selectively constrained and enriched for functional associations

David S.M. Lee [ID] [1], Louis R. Ghanem [ID] [2]* & Yoseph Barash [ID] [1,3]*

G-quadruplex (G4) sequences are abundant in untranslated regions (UTRs) of human messenger RNAs, but their functional importance remains unclear. By integrating multiple sources of genetic and genomic data, we show that putative G-quadruplex forming sequences (pG4) in 5′ and 3′ UTRs are selectively constrained, and enriched for cis-eQTLs and RNA-binding protein (RBP) interactions. Using over 15,000 whole-genome sequences, we find that negative selection acting on central guanines of UTR pG4s is comparable to that of missense variation in protein-coding sequences. At multiple GWAS-implicated SNPs within pG4 UTR sequences, we find robust allelic imbalance in gene expression across diverse tissue contexts in GTEx, suggesting that variants affecting G-quadruplex formation within UTRs may also contribute to phenotypic variation. Our results establish UTR G4s as important cis-regulatory elements and point to a link between disruption of UTR pG4 and disease.

[1] Department of Genetics, Perelman School of Medicine, University of Pennsylvania, Philadelphia, PA 19104, USA. [2] Division of Gastroenterology, Hepatology and Nutrition, Department of Pediatrics, The Children's Hospital of Philadelphia and The University of Pennsylvania Perelman School of Medicine, Philadelphia, PA 19104, USA. [3] Department of Computer and Information Science, School of Engineering, University of Pennsylvania, Philadelphia, PA 19104, USA. *email: GHANEML@email.chop.edu; yosephb@upenn.edu

Most disease-associated genetic variations uncovered through genome-wide association studies reside in noncoding regions of the genome, where they are hypothesized to disrupt regulatory processes important for maintaining normal biological functions[1]. Variants affecting DNA regulatory elements, including transcription factor binding sites and distal-acting transcriptional enhancers are known to contribute significantly to the pathogenesis of multiple diseases[2–4]. In contrast, the importance of functional elements within RNAs is less appreciated on a genome-wide scale.

The 5′ and 3′ untranslated regions (UTRs) of messenger RNAs (mRNAs) encode functional sequences that regulate gene expression through posttranscriptional mechanisms. Known functional elements within UTRs with disease relevance include 3′ UTR polyadenylation signals, which facilitate successful nuclear export of premature mRNAs, and 5′ UTR upstream open reading frames, which modify translation[5,6]. Yet assigning biological significance to the majority of genetic variation within UTRs remains limited by an incomplete understanding of what functional sequences are in UTRs, and how variants in these sequences affect their function. Thus expanding our understanding of regulatory elements in UTRs is important for improving the interpretation of noncoding genetic variation in human health and disease.

Guanine-rich nucleic acid sequences can form noncanonical secondary structures known as G-quadruplexes (G4s) in both DNA and RNA[7]. In contrast to DNA G4s, RNA G4s form more readily in vitro due to their increased thermodynamic stability and reduced steric hindrance[8,9]. Both bioinformatics and experimental approaches have uncovered abundant putative G-quadruplex (pG4) forming regions within the human genome, and it has been observed that G4s are specifically enriched in 5′ and 3′ UTRs of mRNAs, implying that these features are involved in regulating gene expression[10]. Although specific RNA G4s have been associated with diverse biological functions, including mediating translational control[11,12], alternative splicing[13], subcellular localization[14], and RNA stability[15,16], the transcriptome-wide functional importance of UTR G4s has largely been extrapolated from a limited number of experimental studies. The uncertainty over the biological significance of RNA G4s is highlighted by recent findings suggesting that almost all RNA G4s exist in unfolded conformations in cellulo[17]. Thus the question of whether these noncanonical secondary structures are functional remains unresolved.

To address this question, we combine several large-scale genomic and genetic data resources to assess evidence for evolutionary constraint on UTR pG4 sequences in humans, and enrichment for functional associations, including cis-eQTLs and protein-binding sites. We show that UTR pG4 sequences are subjected to heightened selective pressures, and have enrichment for cis-eQTL variants as identified by GTEx, and RNA–protein binding interactions mapped by ENCODE. Taken together, our results support the biological significance of UTR pG4 sequences, and highlight the importance of considering secondary structures in determining biological function in noncoding regions of the genome.

## Results

**pG4 exhibit heightened selective pressure within UTRs.** G-quadruplex (pG4) forming sequences are enriched within UTRs of human mRNAs[18]. If these sequences are functional, they should exhibit patterns of genetic variation consistent with heightened evolutionary constraint. To test this hypothesis, we evaluated the distribution and frequency of single nucleotide variants occurring within UTR pG4 sequences using whole-genome sequencing data from over 15,000 individuals from the public gnomAD release (verison 2.2.1)[19]. We mapped pG4 sequences transcriptome-wide within annotated UTRs using the canonical G4 motif—GGG-{N:1:7}(3)-GGG (Fig. 1a). Consistent with previous UTR G4 mapping efforts[10], we identified 2967 unique protein-coding genes encoding for at least one transcript isoform containing a pG4 sequence within the 5′ UTR, and 2835 protein-coding genes encoding a pG4 sequence within the 3′ UTR (Supplementary Table 1). To further increase the specificity of pG4 sequences, we additionally defined a subset of experimentally supported rG4 sequences (466 in the 5′ UTR, 1743 in the 3′ UTR), consisting of canonical pG4 sequences with evidence of secondary structure formation as determined by biochemical structure mapping approaches[18].

Under the expectation that deleterious variation is continuously removed from the population, we expect allele frequencies for variants affecting UTR pG4 sequences to be skewed toward more rare variation compared to non-pG4 UTR variants, reflecting their greater functional importance[20–22]. Because allele frequencies throughout the genome are affected both by local sequence context, which influences the mutability of a base at a given position, and nearby constrained functional elements that are under linked selection, we compared only single nucleotide variants affecting pG4 G-tracts to non-pG4 G-tracts (three or more Gs) within UTRs belonging to a subset of transcripts whose estimated levels of overall constraint matched our UTR pG4-containing transcripts. This set of comparator transcripts was selected using the upper 90% bound of the observed vs. expected (LOEUF) metric, as published by gnomAD (Supplementary Fig. 2 and see Section "Methods")[19]. This analysis revealed a significant depletion of variants in pG4 and rG4-seq G4s. For rG4 sequences, we found mean allele frequencies were approximately one-third of that compared to non-pG4 G-tracts in constraint-matched transcripts in the 5′ UTR, and 30% lower for the 3′ UTR. For pG4 sequences without direct experimental support, G-tract variant frequency differences were similarly reduced ($P \ll 2.2 \times 10^{-16}$ for 5′ UTR and 3′ UTR; Fisher's exact test). Taken together, this reduction in mean allele frequencies for variants in 5′ and 3′ UTR pG4 sequences relative to those not affecting pG4 sequences is consistent with the effects of negative selection.

To provide a complementary measure of sequence constraint, we assessed the number of polymorphic sites within UTR pG4 sequences compared to non-pG4 sequences. We applied a background model of neutral evolution to produce a distribution for the expected number of polymorphic sites in a given region of the genome under the assumption of neutral selection. This model has been shown to explain a median of 81% of the variability in nucleotide substitution probabilities for noncoding regions of the genome based on the local heptamer context of a given position[23]. Using this model, we partitioned UTR pG4 sequences into G-tracts and intervening gap sequences, and compared the ratio of observed vs. expected polymorphic sites in the European subpopulation of the 1000 genomes project. To additionally control for the possible confounding effects of linked selection driven by nearby constrained coding elements, or differences in sequencing depth across the 1000 Genomes project, we produced an empirical distribution for observed vs. expected substitutions in constraint-matched 5′ and 3′ UTR sequences (Supplementary Fig. 3—see Section "Methods"). Consistent with the observed reduction in variant frequencies across UTR pG4s, we find a significant reduction in the number of observed vs. expected polymorphic sites within UTR pG4 sequences compared to non-pG4 forming regions of the UTR. Relative substitution rates in 5′ and 3′ UTR G-tracts are reduced approximately 30–40% compared to non-pG4 regions of constraint-matched UTRs (permuted $P < 10^{-4}$, for 5′ and 3′ UTR pG4 and rG4)—Fig. 1c.

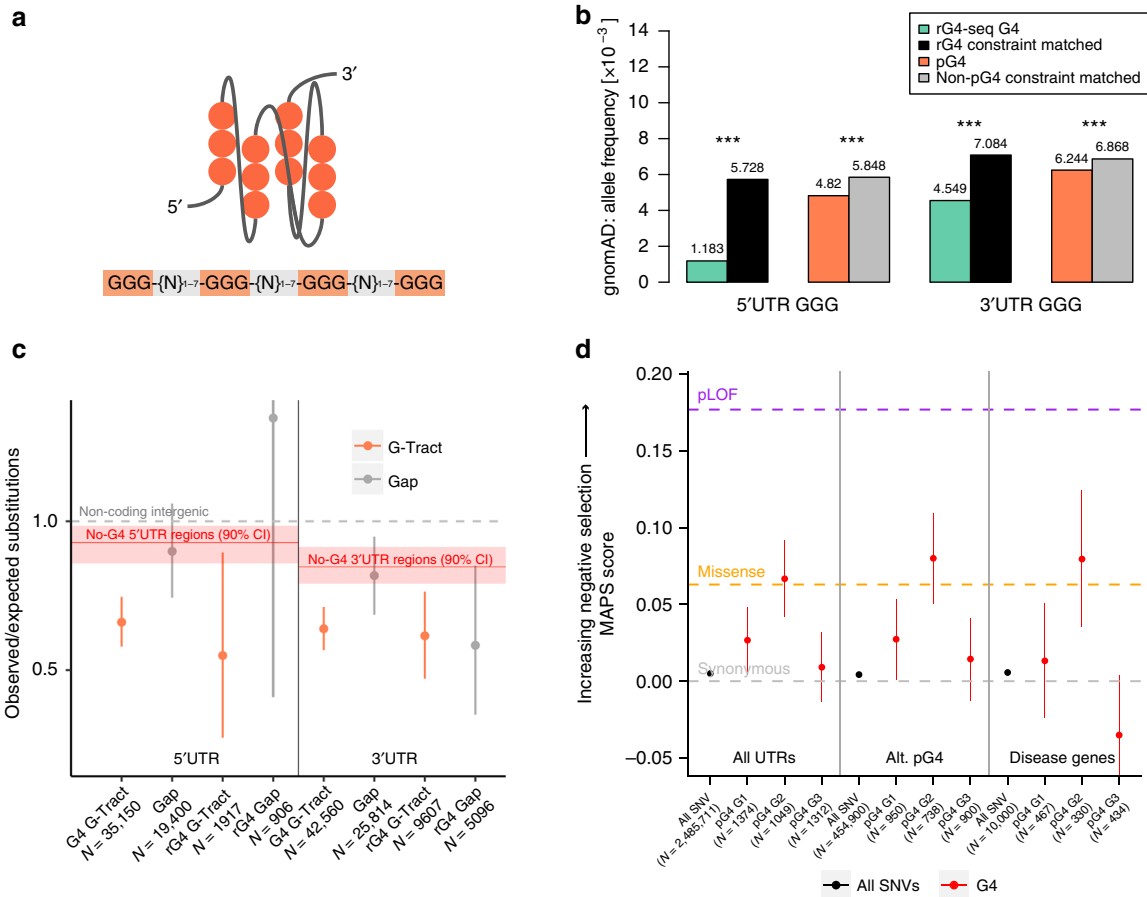

**Fig. 1 UTR pG4 sequences are under heightened selective pressure. a** Schematic depicting a folded RNA parallel G-quadruplex with the accompanying canonical pG4-sequence. **b** Reduction in variant frequencies affecting guanine G-tracts within UTR pG4 forming sequences compared to matched non-pG4 G-tracts by transcript-level constraint. rG4-G-tracts are those within UTR pG4 that have evidence of secondary structure formation by rG4-seq. Asterisks denote $P$ value $<< 2.2 \times 10^{-16}$ by Fisher's exact test. **c** Reduction in the number of observed polymorphic sites compared to expectation in 5′ and 3′ UTR pG4 forming G-tracts using a nucleotide substitution model based on local sequence context (permuted $P < 1 \times 10^{-4}$ in all G-tracts compared to matched non-pG4 UTR sequences). Error bars represent bootstrapped 90% confidence interval for the ratio of observed vs. expected substitutions within each pG4 region. Red line and shaded regions represent the observed vs. expected number of substitutions in non-pG4 UTR sequences matched by transcript-level constraint and 90% confidence intervals, respectively. Gray-dashed line represents an expected vs. the observed ratio of 1:1. **d** Mutability-adjusted proportion of singletons (MAPS) for each set of variants affecting trinucleotide guanines within the meta-pG4 sequence motif. Central position guanines consistently demonstrate the highest MAPS scores (are most constrained) compared to non-pG4 UTR variants (permuted $P < 1 \times 10^{-4}$) across all contexts. Error bars represent the 5% and 95% bootstrap permutations for each variant class. Purple-dashed line, orange dashed line, and gray-dashed line represent MAPS score for Ensembl predicted high-impact coding (predicted loss-of-function), missense, and synonymous mutations respectively. Source data for **b**–**d** are provided as a Source Data file.

In contrast, gap sequences that are not predicted to be important for secondary structure formation in either 5′ or 3′ UTR pG4 contexts are not significantly different from the background UTR estimates, consistent with a pattern of selective pressure in 3′ UTR pG4 sequences that primarily act to maintain the capacity for secondary structure formation across UTR pG4 sequences.

The reduction in allele frequencies, and in the number of polymorphic sites within UTR pG4 sequences, indicate UTR pG4 are under heightened selective pressures compared to non-pG4 UTR regions. To place the degree of selection on UTR pG4s in context, we applied a mutability-adjusted proportion of singletons (MAPS) metric, which measures the relative enrichment for rare variation within a particular class of variants accounting for differences in mutation rates based on local sequence context[24]. A similar approach has been recently used to assess the degree of selective pressure against upstream open reading frame-creating variation within 5′ UTR variants in the gnomAD database[25].

Within the canonical pG4 motif, we predicted that variants affecting the central guanine of each G-tract should be most constrained, since biophysical studies of G4 stability have shown that mutations affecting the central tetrad (second guanine of each trinucleotide guanine repeat) are most detrimental to secondary structure stability[26]. To remove the ambiguity of which specific guanines are involved in secondary structure formation when more than three guanines form a pG4 G-tract, we focused only on single nucleotide variants within trinucleotide G-tracts ($n = 3137$). By examining variation across each pG4 G-tract, we found central guanine positions within UTR pG4 G-tracts are consistently enriched for singletons (one sequenced variant in gnomAD whole genomes) compared to non-pG4 UTR variants (Fig. 1d, permuted $P < 10^{-4}$). Notably, non-pG4 UTR variants reflected a similar degree of constraint as synonymous coding variants, while central position guanines exhibit a similar degree of selective pressure as missense variation in protein-coding regions of the genome. Interestingly, the most

proximal and distal 5′ and 3′ guanine of each trinucleotide pG4 G-tract demonstrated significantly less enrichment of singleton variants within gnomAD across gene classes compared to central guanine positions as determined by permutation testing (P values = 0.0237 and 0.0022, respectively—Supplementary Fig. 4). This result suggests these positions are under less selective pressure compared to central positions, perhaps because mutations in these positions can preserve the potential for RNA to form noncanonical G4 2-quartets[18]. Finally, to provide additional control for our sequence context-derived mutability rates, we compared the MAPS metric for UTR pG4 G-tracts to UTR trinucleotide G- and C-runs not involved in pG4 formation. Although these non-pG4 G- and C-tracts exhibit modest enrichment for rare variation at the central position, there is a significantly greater enrichment in singletons at the central position of the UTR pG4 G-tract compared to non-pG4-forming contexts (Supplementary Fig. 4—permuted P = 0.0195). Thus, the excess rare variation is specific to the guanine within pG4 G-tracts most important for maintaining G4 secondary structure.

**Most pG4 motifs in UTRs are isoform-restricted**. Many functional UTR elements, including upstream open reading frames (uORFs), AU-rich elements, and microRNA-binding sites are frequently included in alternative 5′ or 3′ UTR isoforms of the same gene[27,28]. Alternative UTR inclusion is hypothesized to significantly diversify the number of possible posttranscriptional regulatory interactions for a given gene[29]. Given the observed constraint over UTR pG4 sequences, we hypothesized that UTR pG4 sequences should also exhibit patterns of alternative inclusion or exclusion.

To evaluate the extent of alternative UTR pG4 inclusion, we mapped UTR pG4s to protein-coding transcripts for each gene in the Ensembl transcriptome database. Genes were considered to produce constitutive UTR pG4 sequences when all annotated protein-coding transcript isoforms contained at least one pG4, or alternative UTR pG4 sequences if at least one transcript isoform lacked the pG4 sequence. Most constitutive pG4 genes were found to express UTRs with identical pG4s across all transcript isoforms, however, 36 of 620 5′ UTR and 75 of 1275 3′ UTR constitutive pG4 genes produced transcript isoforms with nonidentical pG4 sequences. For this subset of nonidentical pG4-transcript isoforms, approximately one-third differ by the addition/subtraction of pG4 motifs (13/36 for 5′ UTR, 20/75 for 3′ UTR). Strikingly, we found that over half of all genes producing UTR pG4 transcripts also encoded for alternative UTRs lacking pG4 motifs (2254 genes with 5′ UTR pG4 motifs and 1425 genes with 3′ UTR pG4 motifs—Fig. 2a). Indeed, of the 5235 total UTR pG4-containing genes, 3395 exhibited either alternative 5′ or 3′ UTR pG4 inclusion, and 284 produced UTRs with both alternative 5′ and 3′ pG4s. This distribution of alternative and constitutive pG4 genes for each UTR context was found to be highly significant through permutation testing (P value < 0.0001 for 5′ and 3′ UTRs). Moreover, MAPS scores for alternative UTR pG4 indicate that their second guanine position is under a similar degree of constraint as for all UTR pG4, and is comparable to that of missense variations for the set of alternative pG4s found in genes with any disease association in ClinVar (https://www.ncbi.nlm.nih.gov/clinvar/) (Fig. 1d). As is the case for all UTR pG4, this second guanine position was significantly more enriched for rare variation compared to either the 5′ or 3′ guanine (permuted P value = 0.0138 and 0.004, respectively). Constitutive UTR pG4 sequences, in contrast, do not show a similar pattern of selective constraint acting on the second G-tract guanine, possibly because these sequences tend to be under less stringent selective pressures, or because we are underpowered to detect significant enrichment in rare variation. Notably the MAPS metric for the central G-position of alternative pG4 sequence G-tracts remained significantly higher than matched, non-pG4 G-tracts (permuted P = 0.0124—see Supplementary Fig. 4 for comparison of constitutive pG4 G-tracts and other pG4 gene sets).

We next asked whether the expression of alternative pG4 isoforms tend to be restricted or shared across different tissue contexts. Using transcript-isoform expression data across 45 different tissues from GTEx, we find that many tissues appear to express both pG4 and non-pG4 transcripts simultaneously (Fig. 2b). Notably, this simultaneous expression of both pG4 and non-pG4 isoforms also occurs in single-cell contexts (lymphocytes and fibroblasts), demonstrating that this effect is not due to cellular heterogeneity in bulk tissue samples. Thus most UTR pG4-encoding genes express alternative isoforms which lack pG4 sequences, and the simultaneous expression of both pG4-isoforms and non-pG4 isoforms is widespread across multiple tissue and cellular contexts.

To explore the functional associations of alternative UTR pG4 genes we performed a gene ontology analysis. We find that these genes are frequently involved in dynamic intracellular processes, including signal transduction, cellular responses to stress, and metabolic regulation (Fig. 2c—Supplementary Data 2). In contrast, constitutive pG4 genes showed enrichment for biological processes associated with the activation of gene expression in discrete temporal stages, including those involved in tissue development, pattern specification, and cellular differentiation (Supplementary Data 2). These observations, coupled with our finding that many tissues simultaneously express both pG4 and non-pG4 isoforms of the same gene, suggests that isoform-switching between pG4-containing or non-pG4 transcripts may facilitate dynamic cellular responses to external stimuli. More broadly, our results demonstrate considerable variation in alternative pG4 inclusion within UTRs across multiple tissue contexts, and suggest that the relative abundance of pG4 and non-pG4 UTRs may be dynamically regulated within a given tissue.

**pG4 motifs in the 5′ and 3′ UTR are enriched for *cis*-eQTLs**. We next evaluated the potential regulatory consequences associated with mutations affecting UTR pG4s, hypothesizing that variants affecting pG4 sequences might also be more likely to be associated with changes in gene expression. To test this hypothesis, we compared the proportion of annotated *cis*-eQTLs vs. non-eQTL SNPs identified by GTEx across pG4 and non-pG4 regions of the UTR, finding significant enrichment for either nominally significant or lead eQTL variants (lowest P value variant) in 5′ and 3′ UTR pG4 sequences compared to non-pG4 regions of UTRs (Fig. 3a—statistics on enrichment in Supplementary Table 1). Notably, we continue to observe an enrichment of *cis*-eQTL variants in UTR pG4 sequences using a reduced set of putatively causal *cis*-eQTLs[30], suggesting that disruption of UTR pG4 sequences may cause changes in posttranscriptional regulation.

We next explored the direction of gene expression changes for UTR pG4 *cis*-eQTLs, considering all variant-tissue effects separately for each significant variant-tissue interaction. We hypothesized that variants affecting pG4 G-tracts are more likely to disrupt the structural integrity of the RNA G4s, and thus might influence gene expression differently than variants affecting gap (non-G-tract) sequences within pG4 motifs. Since the magnitude of normalized effect-size estimates in GTEx has no direct biological interpretation, we compared differences in the direction of variant effects across pG4 and non-pG4 sequences.

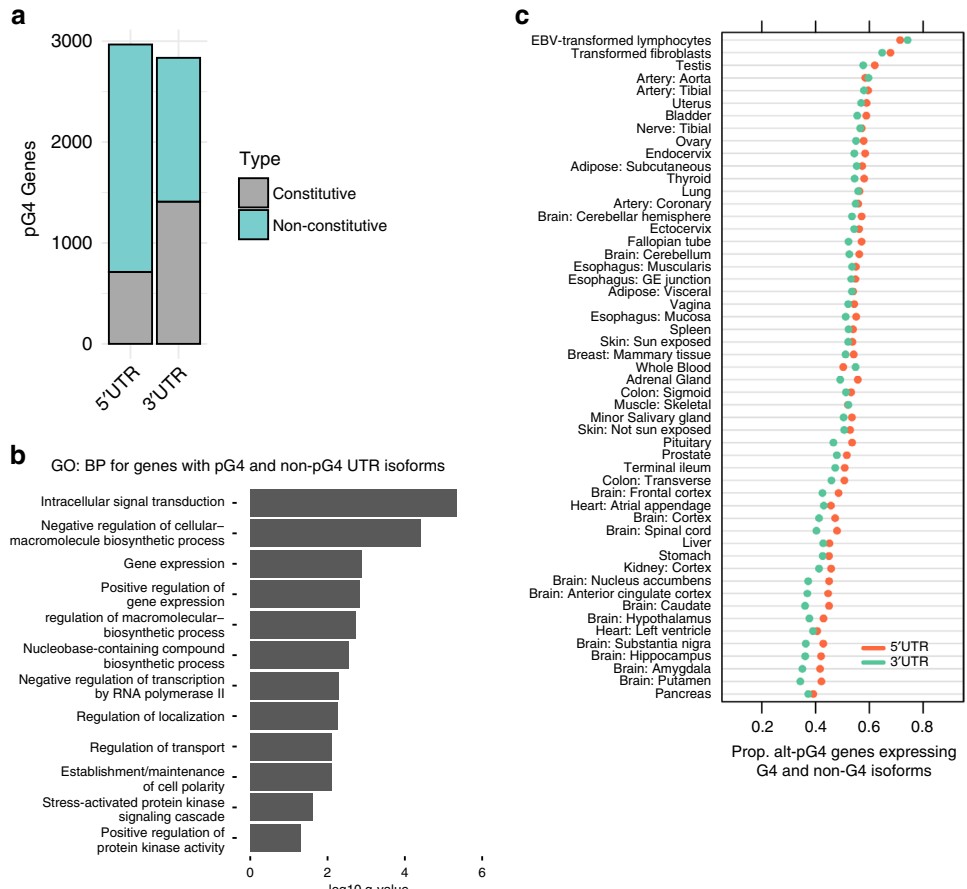

**Fig. 2 Isoforms with G4 in UTR distribution and usage. a** Most genes producing mRNA transcripts with UTR pG4 sequences also produce alternative isoforms that lack UTR pG4s (non-constitutive). **b** For the subset of genes that produce UTRs with alternative pG4 inclusion, both pG4-containing and non-pG4 isoforms are frequently expressed simultaneously. Median expression (TPM) of each pG4-transcript or non-pG4 transcript was assessed for each tissue context. Transcripts were considered as expressed if their median TPM measurement exceeded one TPM for each tissue context considered. The proportion of pG4 genes expressing both pG4 isoforms, and non-pG4 isoforms was then compared for each tissue. **c** Overrepresented biological processes for protein-coding genes producing both pG4 and non-pG4 5′ or 3′ UTR isoforms ($n = 3148$)—see Supplementary Data 1 for the full list. GO-term enrichment was performed using PantherDB[55] and enrichment was determined by meeting a Benjamini–Hochberg adjusted $P$ value cutoff of 0.05 by Fisher's exact test. Source data are provided as a Source Data file.

As expected, UTR variants in non-pG4 regions are not significantly biased towards increasing or decreasing gene expression, regardless of whether the mutation affected a G-tract, or non-pG4 G-tract nucleotide. In contrast, mutations affecting structurally important pG4 G-tracts in the 3′ UTR tend to increase mRNA expression compared to non-G-tract bases (OR 1.75, 95% CI: 1.34–2.30, $P < 3.0^{-5}$)—Fig. 3b. This relationship for the 5′ UTR was not observed. Given the role of the 3′ UTR in mediating mRNA stability, the tendency for G-tract base mutations to increase gene expression suggests the involvement of 3′ UTR G4s in decreasing mRNA stability.

**RNA–protein binding sites are enriched over UTR pG4 regions.** Transcriptome-wide RNA structure mapping studies have suggested that most RNA G4 are unfolded in eukaryotes, but not in prokaryotes, leading to the hypothesis that intracellular factors bind RNA G4s to maintain their unfolded state in cellulo[17]. To gain insights into regulatory mechanisms mediating pG4 effects on gene expression, we investigated the propensity of protein-binding sites to overlap UTR pG4s by comparing the proportion of UTR pG4 sequences overlapped by RNA-binding protein (RBP) binding sites published by ENCODE to non-pG4 forming regions of the UTR[31]. This data consists of cross-linking

immunoprecipitation sequencing (CLIP-seq) peaks, called from K562 or HepG2 cell lines for over 150 RBPs, containing at least one highly reproducible (IDR = 1000)[32] binding peak within the 5′ or 3′ UTR. When compared to non-pG4 regions of the UTR, the frequency of overlap between unique (nonoverlapping) RBP-binding sites and pG4 sequences was almost 6-fold ($P \ll 2.2 \times 10^{-16}$, chi-square test) higher compared to non-pG4 sequences in the 5′ UTR (Fig. 3c). Enrichment of RBP-binding locations over pG4 sequences within the 3′ UTR was markedly higher (14-fold, $P \ll 2.2 \times 10^{-16}$, chi-square test). Given the enrichment within UTR pG4s for *cis*-eQTLs and protein-binding sites, we tested for significant colocalization between these two features in pG4s. Taking the subset of pG4 regions overlapped by any protein-binding sites, we examined the density of *cis*-eQTLs in UTR pG4 regions also overlapping CLIP-seq peaks. When all nominally significant *cis*-eQTLs are considered, we observe a significant enrichment of *cis*-eQTLs in the 3′ UTR that are also protein-binding sites (Fig. 3a), indicating that variation in 3′ UTR pG4 sequences may influence gene expression through changing RNA–protein interactions.

Given the observed association between protein-binding sites and pG4 sequences, we next asked whether specific proteins' binding sites are enriched for pG4s. For each protein, we determined the proportion of protein-specific binding sites

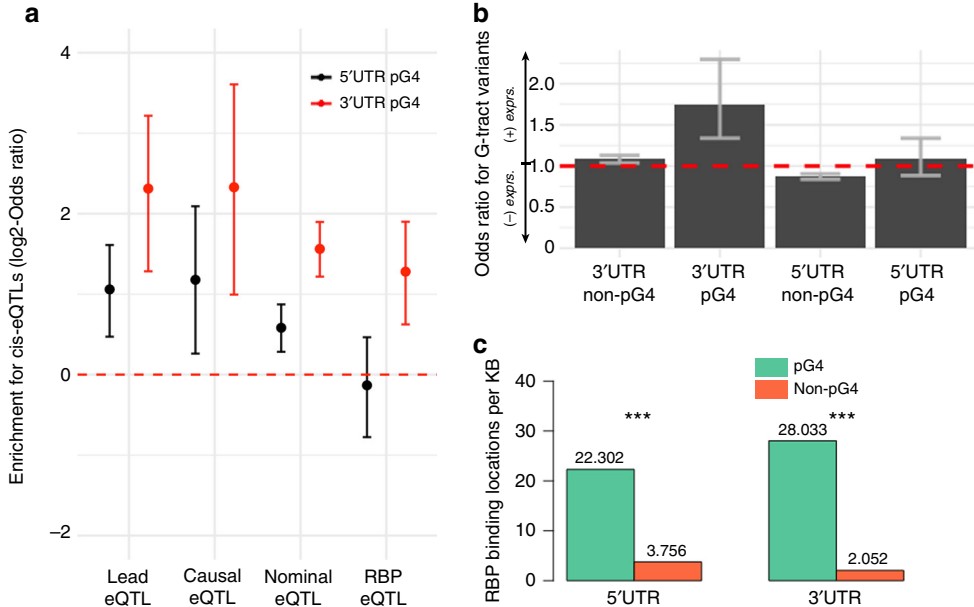

**Fig. 3 Enrichment for functional associations within UTR pG4 sequences. a** GTEx *cis*-eQTLs are enriched within UTR pG4 relative to the number of tested (non-eQTL) SNPs when comparing lead SNPs, high-confidence causal, nominally significant, and nominally significant in RBP-binding sites in matched UTR regions (see Supplementary Table 3 for enrichment statistics) Fisher's exact test. Error bars represent the 95% confidence interval for the odds ratio. **b** Odds ratio for a *cis*-eQTL increasing gene expression across all *cis*-eQTL-tissue effects ($n = 379,441$, $P$ value $< 2e{-}16$, Fisher's exact test), where the variant affects a pG4 G-tract compared to those affecting gap sequences. Error bars represent the 95% confidence interval for the odds ratio. **c** Density of RBP-binding sites per kilobase of pG4 sequence compared to non-pG4 regions of the UTR ($P$ value $\ll 2.2 \times 10^{-16}$, chi-square test). Source data are provided as a Source Data file.

containing pG4 sequences, against the total background rate of all CLIP-seq binding sites containing pG4 sequences. To determine a significant overrepresentation of pG4 sequences within a given protein's binding sites, we performed a hypergeometric test against the null hypothesis that there is no overrepresentation of pG4-binding sites within the set of a given protein's binding sites —Fig. 4a. This analysis revealed enrichment for proteins that have been implicated in RNA G4 binding (GRSF1, FUS), and those that, to our knowledge, have not previously been associated with RNA G4 structures (PRPF4, GTF2F1, and CSTF2T). GRSF1 is a cytoplasmic protein involved in viral mRNA translation, and has recently been shown to play a role in the degradation of G4-containing RNAs in mitochondria[33,34]. Other proteins with significant enrichment for pG4 binding, include those involved in mitochondrial processes (FASTKD2), transcriptional activation (GTF2F1), mRNA transport (FAM120A), mRNA degradation (XRN2, UPF1), in addition to several proteins implicated in RNA polyadenylation and splicing (CSTF2T, PRPF4, RBFOX2), and surprisingly, micro-RNA (miRNA) biogenesis (DCGR8, DROSHA). Interestingly, proteins demonstrating a preference for binding UTR pG4 sequences tend to bind both 5′ and 3′ UTR contexts, with 14 out of 20 proteins' binding peaks showing enrichment for overlap over 5′ and 3′ pG4 sequences in HepG2, and 17 out of 25 for K562 independently (Supplementary Table 2). Taken together, these data suggest that RBP binding is enriched in UTRs over pG4 sequences, and that RBP–pG4 interactions may regulate gene expression.

An analysis of gene expression changes with sh-RNA knockdown for the majority of pG4-enriched binding proteins showed genes containing pG4 in either the 5′ or 3′ UTR are much more likely to be significantly differentially expressed compared to non-pG4 genes (Supplementary Fig. 5a, b). Approximately, one-third of the proteins exhibiting a binding preference for UTR pG4 change the expression of pG4-containing genes concordantly

across K562 and HepG2 cells (GTF2F1, FASTKD2, UPF1, NONO, GRSF1, NCBP2, AKAP8L, DDX6, FKBP4, TAF15, and LARP4). Of these 11 RBPs, knockdown of 8 tends to decrease expression of UTR pG4 genes (GTF2F1, UPF1, NONO, GRSF1, NCBP2, AKAP8L, DDX6, and LARP4), while knockdown of three (FASTKD2, FKBP4, and TAF15) tends to increase their expression, suggesting that most of the proteins enriched for pG4 binding tend to increase, or stabilize RNA expression rather than facilitate their degradation. This result is consistent with our finding that *cis*-eQTLs affecting 3′ UTR pG4 sequences are more frequently associated with decreasing gene expression.

Finally, to explore the potential existence of posttranscriptional regulatory networks relying on shared RNA G4-protein interactions, we tested for a significant overlap in pG4-containing transcripts targeted by each protein enriched for pG4-binding interactions. Taking the set of 31 proteins with significant overrepresentation for pG4 binding (Bonferroni-corrected $P < 0.001$) and at least 20 unique pG4-binding sites in HepG2 or K562, we assessed overlaps between the various proteins' pG4 gene targets (Fig. 4c, d). We found low overlap of targets in helicases that have been hypothesized to bind RNA G4s frequently, such as DDX6, DDX51, and DDX52. In contrast, we find a subset of G4-binding proteins sharing a significant degree of overlap in G4-gene targets, including FASTKD2, FAM120A, CSTF2T, PRPF4, and GTF2F1, none of which have been shown to bind RNA G4 structures previously. These data point to possible mechanisms of gene control relying on the shared interactions of these proteins with their respective RNA targets. Indeed, assessing the functional associations of 133 pG4 genes sharing at least 3 out of 5 protein-binding interactions from this module revealed enrichment for genes involved in viral process (GO:0016032, FDR-adjusted $P = 0.0268$, Supplementary Data 3), suggesting that these genes and putative pG4-binding proteins, may be involved in mediating host–viral interactions within the cell.

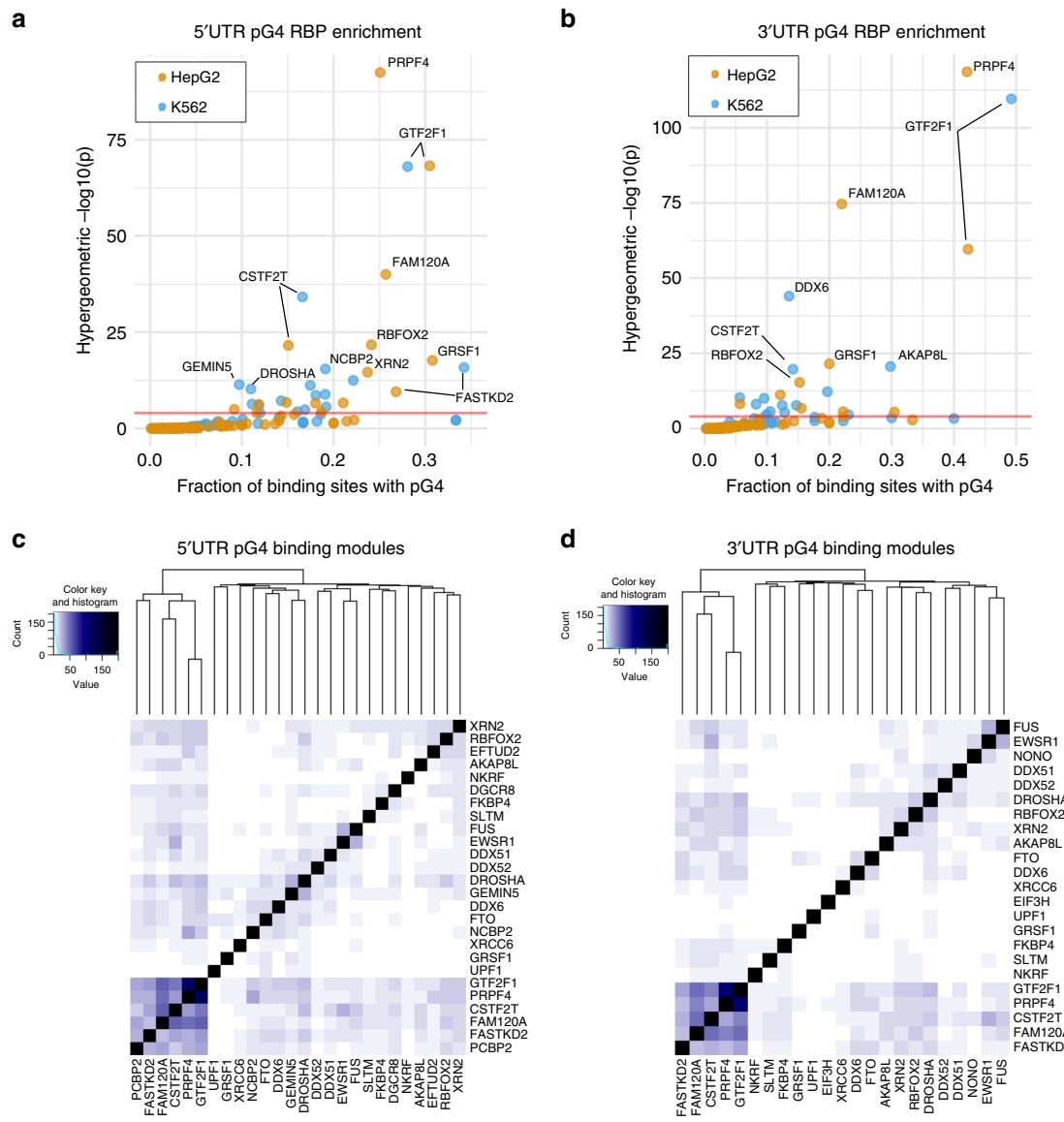

**Fig. 4 Enrichment of specific protein–pG4 binding sites using CLIP-seq data from ENCODE. a**, **b** Enrichment of specific proteins over pG4-binding sites within the 5′ UTR (left) and 3′ UTR (right)—red line corresponds to $P = 0.0001$ (hypergeometric test). **c**, **d** Heatmaps depicting the significance of overlap (hypergeometric −log $P$ value) in pG4 gene targets for proteins found to bind pG4 sequences preferentially. Source data are provided as a Source Data file.

**3′ UTR pG4 in disease-causing genes are enriched for variants.** Multiple studies assessing evolutionary constraints in protein-coding regions of the human genome have shown that regions depleted of genetic variation are also enriched for pathogenic variation[35–37]. Under the principle that purifying selection removes deleterious variants from the genome to produce regions depleted of genetic variation, we expect UTR pG4s should also be enriched for pathogenic variation. Since pathogenic variants in ClinVar are overwhelmingly annotated in protein-coding regions of the genome, we are underpowered to test for a direct association between the set of annotated pathogenic variants and UTR pG4 sequences. Instead, we asked whether potentially pathogenic variation in ClinVar is enriched within UTR pG4 sequences in known disease-associated genes. To test this hypothesis, we mapped all single nucleotide variants annotated in the most recent release of the ClinVar database[38] (April, 2019) across UTRs, and compared their relative density in pG4 vs. non-pG4 sequences in disease-associated genes, excluding variants with an annotation of Benign or Likely_benign. We defined the set of disease-associated genes as any gene with at least

one variant having an annotated as Pathogenic or Likely Pathogenic in ClinVar. To maximize our power for this analysis, we expanded our set of rG4-seq G4s to include all noncanonical G4-forming sequences mapped and reported by rG4-seq in HeLa cells[18]. We found modest enrichment for variation in 3′ UTR pG4 sequences, rG4 3′ UTR sequences, and a notable enrichment in 3′ UTR pG4 sequences within annotated RBP-binding sites from ENCODE in disease-associated genes compared to non-pG4 forming regions of the 3′ UTR—Fig. 5a (All pG4: OR = 1.51, 95% CI: 1.20–1.88, $P < 0.0005$, rG4-seq pG4: OR = 1.18, 95% CI: 0.98–1.42, $P = 0.067$, RBP pG4: OR = 6.01, 95% CI: 3.87–8.91, $P < 5e^{-12}$ Supplementary Data 4 —Fisher's exact test). In contrast, there was only evidence for enrichment of variants in the 5′ UTR rG4 sites (OR = 2.32, 95% CI: 1.93–2.78, $P < 2.25 \times 10^{-16}$ Fisher's exact test), but not pG4 or pG4-RBP overlap regions. It is important to note though that the above statistical test only contrasts relative enrichment of putative pathogenic variants in pG4 vs. non-pG4 UTR sequences. Thus, the lack of such relative enrichment in the 5′ UTR may reflect the generally greater density of other functional elements within 5′ UTR sequences.

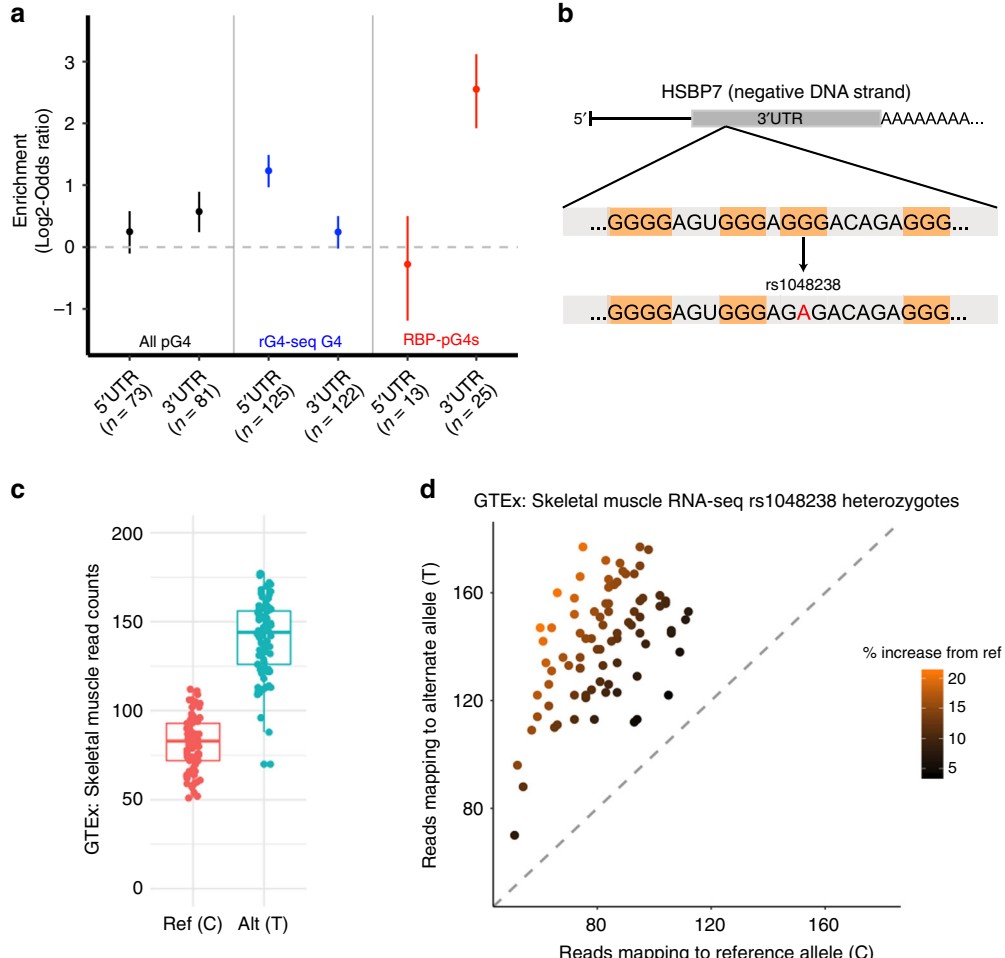

**Fig. 5 UTR pG4 sequences are enriched for known pathogenic, and putative disease-associated genetic variants. a** Annotated variants within ClinVar disease-associated genes occur with greater frequency in UTR pG4 sequences compared to non-pG4 UTR regions in the 3′ UTR across multiple G4 subsets (error bars represent the 95% confidence interval). **b** rs108348 maps to a 3′ UTR pG4 G-tract guanine within the primary HSPB7 transcript, which is encoded on the negative DNA strand. The SNP disrupts the canonical G4 sequence motif by causing a G to A mutation in the RNA transcript. **c, d** WASP-mapping of allele-specific reads in 84 GTEx skeletal muscle samples reveals significant allelic imbalance favoring expression of the alternative allele ($P$ value < 1 x $10^{-100}$, likelihood ratio test). Boxplot in **c** represents median and 1.5 times the interquartile range of WASP-aligned RNA-seq reads aligning to the ference (red) or alternative (blue) allele. Source data are provided as a Source Data file.

In conclusion, these data imply that disease-associated noncoding variation may be enriched in 3′ UTR pG4 regions.

Finally, we tested for enrichment of common variants that have been associated with disease phenotypes using annotations available in the NIH genome-wide association study (GWAS) Catalog. There were not enough GWAS-associated lead variants within UTR pG4 regions to detect enrichment (seven variants in 5′ UTR pG4, four in the 3′ UTR pG4—Supplementary Data 5). However, given the enrichment for *cis*-eQTLs in UTR pG4, we hypothesized that disruption of UTR pG4 sequences could affect posttranscriptional mechanisms regulating gene expression, thus providing a potential mechanistic link between GWAS variants and their observed phenotypes. To test this hypothesis, we assessed evidence of allelic imbalance at select GWAS SNPs either falling within a UTR pG4 region, or in high LD with a common SNP ($r$-squared > 0.85 in the GBR population of 1KG) falling within a UTR pG4 in GTEx. Despite being limited by the number of heterozygous individuals in GTEx with matched whole-genome sequencing available, our analysis uncovered several proxy SNPs in high LD with GWAS tag-SNPs (Supplementary Fig. 7, Supplementary Table 6), and one GWAS lead variant exhibiting evidence of significant allelic imbalance. The lead

GWAS variant, rs1048238 is a common SNP within the 3′ UTR of *HSPB7*, a chaperone protein that is highly expressed in heart and skeletal muscle and has been associated with hypertension in a recent GWAS[39,40] (Fig. 5b). We found that rs1048238 exhibited a substantial imbalance of reads mapping to the alternative allele in 84 heterozygous individuals, even after correcting for read-mapping biases using WASP-filtering[41]. Taken together, these results demonstrate that the predicted pG4-disrupting variant is associated with increased expression of the alternative allele at this locus (Fig. 5c, d). This association is consistent with our previous observations from transcriptome-wide mapping of pG4 eQTL showing that 3′ UTR pG4 eQTLs tend to increase gene expression (Fig. 3b), and suggest that the impact of these variants on gene expression are responsible for their respective GWAS associations. Additional variants exhibiting allelic imbalance and their respective disease associations are shown in Supplementary 5 and reported in Supplementary Table 7.

## Discussion
We have applied a deep catalog of human genetic variation to assess evolutionary pressures over putative G-quadruplex forming

sequences within 5′ and 3′ UTRs. We hypothesized that if these regions are functionally important they should be depleted of variation. Supporting this hypothesis, we show that variation within UTR pG4 sequences is reduced compared to non-pG4 UTR regions using a local sequence context based substitution model. Moreover, our analysis of positional constraint within the meta-pG4 motif reveals selective pressures acting on central guanines of each trinucleotide G-tract comparable to that of missense mutations in protein-coding regions of the genome. These findings are consistent with in vitro biophysical studies of DNA G-quadruplex stability, which have shown that central position substitutions are most destabilizing, and consequently were predicted to be the most deleterious for native biological functions of G4s[26,42,43]. Interestingly, we find that noncentral guanines appear less constrained compared to central positions—possibly because mutations at these positions may preserve the potential for RNA to form noncanonical G4 2-quartets. Indeed, these G4 2-quartets have been estimated to account for one-fourth to two-third of all RNA G4 structures observed by transcriptome-wide rG4-seq in HeLa cells[18].

We also uncover a greater proportion of cis-eQTLs mapping to pG4 regions compared to non-pG4 sequences within both 5′ and 3′ UTRs. Our analysis of nominally significant cis-eQTL enrichment in UTR pG4 sequences may be confounded by the presence of linked SNPs that reach nominal significance because of their proximity to causal eQTL SNPs, however, this likely deflates our estimates of enrichment in UTR pG4 sequences because the relatively smaller size of pG4 motifs (15–33nt) makes multiple linked nominally significant cis-eQTLs more likely to occur along the length of non-pG4 UTR regions. Nevertheless, the enrichment of cis-eQTLs within UTR pG4 remain unchanged when we limit each UTR pG4 feature to contain at most one nominally significant cis-eQTL SNP (Supplementary Table 1).

Using CLIP-seq data for over 150 proteins published by ENCODE, we find 15 proteins whose binding sites are enriched for pG4 sequences across two cell lines, and identify regulatory modules associating a set of RNA-binding proteins, including FAM120A, FASTKD2, and CSTF2T, with pG4 gene targets involved in viral mRNA expression. Indeed, several examples of viral hi-jacking of eukaryotic RBPs have been reported in the literature[44,45], and G4-forming sequences have been found to occur commonly in multiple viral genomes[46]. This, coupled with the observation that RNA G4s appear to be universally depleted within prokaryotic transcriptomes[17], suggests that viruses might rely on G4s as a mechanism for co-opting host cell machinery involved in gene expression and RNA regulation.

There are three primary limitations to the current study. First, we applied a text-based approach toward identifying regions of putative G-quadruplex formation within RNA UTRs. Although this approach has been commonly employed in previous work[10,18], there exists a considerable literature regarding possible variations to the canonical G-quadruplex forming sequence and methods that capture more variable motif definitions[47–49]. Given the comparably limited evidence that many of these alternative G-quadruplex sequences form readily in cellulo we used a more stringent motif definition, but alternative G4 sequences will have been missed in our analysis. The modest enrichment in singletons at the central position of trinucleotide G- and C-tracts not matching our canonical pG4 sequence motif is consistent with this possibility. Thus, our assessment of sequence constraint and functional enrichment within UTR G4-forming regions is likely incomplete. Secondly, although we have uncovered evidence suggesting G4 secondary structure formation is constrained, whether these pG4s form secondary structures in vivo remains unclear. Finally, our assessment of selective pressures acting across UTR pG4 sequences using the MAPS metric is limited in

power by low variant numbers. Nevertheless, we report multiple lines of evidence supporting the biological importance of putative secondary structure-forming G-quadruplexes within UTRs. Although RNA UTRs represent only a small fraction of the noncoding genome, they are core components involved in mediating posttranscriptional regulation of gene expression. Ultimately we hope this work will motivate researchers to consider G4s and other RNA elements in UTRs when assessing the possible impact of genetic variations in human health and disease.

## Methods

**Identification of UTR G-quadruplex Sequences.** Annotated UTR sequences and genomic coordinates were downloaded using biomaRt[50,51] for Ensembl Transcript Database version 75 for all protein-coding transcripts. Putative G-quadruplex forming sequences (pG4) were identified using this set of annotated UTRs by performing a pattern-matching text search to identify regions of UTRs matching the canonical G-quadruplex pattern as a regular expression: (G:3+)-{N:1–7}(3)-(G:3+). Genomic coordinates for pG4 sequences within UTRs were obtained using a custom python script, and cross-referenced with annotated protein-coding regions of the genome from the Ensembl Transcript Database to remove overlaps between annotated UTRs and coding sequences using human genome hg19 coordinates. The identified set of 5′ and 3′ pG4 sequences, and corresponding genomic coordinates were used for all downstream analysis. This approach yielded a total of 5235 protein-coding genes harboring a UTR pG4 sequence, with 2967 genes having a 5′ UTR pG4, and 2835 genes having a 3′ UTR pG4. Of the 5235 genes with either a 5′ or 3′ UTR pG4, 567 have both.

A second set of pG4 sequences with evidence of secondary structure formation was defined by overlapping pG4 motifs identified transcriptome-wide with published in vitro rG4-seq annotations by using the K + PDS conditions[18]. Only the subset of rG4-seq G4s matching the canonical pG4 motif were used in this analysis. For the set of rG4 sequences, we find 243 protein-coding genes encoding a 5′ UTR rG4, and 803 genes encoding a 3′ UTR rG4. Of these rG4 genes, 16 have 5′ and 3′ UTR pG4-encoding transcript isoforms.

**Constraint analysis.** Variants from gnomAD release 2.2.1 were obtained from (URL: https://gnomad.broadinstitute.org/downloads) and filtered to exclude those marked with segmental duplication, low complexity regions, and decoy flags, in addition to those variants whose true positive probability as determined by a random forest model trained in gnomAD did not exceed 40%[19]. As an additional requirement, only those variants where the total observed allele number was at least 80% of the maximum number of sequenced alleles was considered to control for differences in sequencing depth in the gnomAD WGS dataset. The remaining set of high-confidence variants was overlapped with genomic coordinates for UTR pG4, non-pG4, and CDS regions, using bedtools2 (version 2.27.1) intersect with the -u and -b flags.

The transcript constraint-table from gnomAD release 2.2.1 (URL: https://gnomad.broadinstitute.org/downloads) was used to randomly select a matching set of transcript-level constraint-matched non-pG4 UTR sequences based on the gnomAD observed/expected metric for the 5′ UTR and 3′ UTR separately. Specifically, transcript constraint was matched between pG4 and non-pG4 forming sequences using the observed vs. expected ratio of loss of function variants metric (LOEUF) provided for each transcript by gnomAD.

The fraction of variants per sequenced allele across UTR regions were computed as the fraction of the observed allele count vs. observed allele number. The distribution of frequencies for variants mapping to each UTR region was extracted from the gnomAD summary variant call files directly. P values for difference between the expected number of variants per sequenced allele across genomic regions were calculated using a two-sided Fisher exact test. Only variants that did not overlap annotated coding regions of the transcriptome were compared to ensure that UTRs overlapping coding regions of other transcript isoforms were excluded. All statistical tests were conducted for 5′ UTR and 3′ UTR features separately.

For positional constraint analysis, we applied the MAPS metric[24] for each nucleotide position across all trinucleotide G-tracts with G4-forming capacity, as defined by our bioinformatic analysis. We developed a MAPS model using custom code based on a previously published MAPS model (https://github.com/pjshort/dddMAPS) with the addition of adjusting for methylation levels at variant positions[19]. We divide variants by median estimated methylation levels across 37 tissues at CpG sites into none/low (<0.2), intermediate (0.2–0.6), or high (0.6<) bins for which separate methylation-adjusted mutation rates were available. Our model was trained by regressing the observed proportion of singleton synonymous variants for each trinucleotide context within protein-coding regions of the genome on mutation rates for each trinucleotide context (methylation-adjustment was performed only at CpG dinucleotides) derived from intergenic noncoding regions of the genome[19,52]. All variants used in this analysis, including synonymous variants used for training the model, were subject to the same filtering requirements as used in the analysis of allele frequencies (random forest true positive probability exceeding 40%, and total observed allele number was at least

80% of the maximum number of sequenced alleles). To control for ambiguity regarding which specific guanines within each G-tract are involved in pG4 formation for G-tracts having more than three guanines, we considered only variants within trinucleotide G-tracts. MAPS values were also determined for the set of variants with a VEP consequence of missense, or those variants predicted to cause a loss of function (pLoF) in gnomAD to provide context for the different degrees of purifying selection acting over a set of variants. pLoF variants were defined as those annotated with Ensembl predictions for having a high impact and includes transcript_ablation, splice_acceptor_variant, splice_donor_variant, stop_gained, frameshift_variant, stop_lost, and start_lost terms. In our assessment of positional constraint within the meta-pG4 sequence consisting of only trinucleotide G-tracts, we calculated MAPS for four categories of pG4 variants: (1) all genes, (2) genes with at least one transcript falling in the upper one-third of transcripts that are most intolerant to loss of function mutations (as determined by the gnomAD o/e metric), (3) alternative pG4 genes, and (4) alternative pG4 disease-associated genes extracted from ClinVar database (April 2019 release). Permutation $P$ values were obtained by performing 10,000 bootstraps for each set of pG4 variants in gnomAD with replacement to produce a distribution of MAPS score for each variant context; and then comparing these distributions to a matched set of resampled MAPS scores using either all UTR variants, or position-matched non-pG4 GGG/CCC trinucleotide variants to determine the proportion of bootstrapped samples whose MAPS score of pG4 regions exceeded the matched non-pG4 variant set.

Posterior substitution probabilities for noncoding regions of the genome based on local heptameric sequence contexts were obtained from a published model[23] based on the Phase 1 release of the 1000 Genomes Project. Cumulative substitution probabilities for each of the possible mutations within a heptamer context (e.g., $A \rightarrow C$, $A \rightarrow T$, $A \rightarrow G$) were calculated by summing over all nucleotide substitution probabilities for a given heptamer context. To produce a null distribution of observed/expected number of substitutions for non-pG4 regions of the UTR, we randomly sampled 5000, 25 nucleotide UTR regions from constraint-matched transcripts, and 10,000 times to generate a null distribution. Specifically, constraint-matched non-pG4 transcripts were divided into heptamers using a sliding window across the entire region, and substitution probabilities based on heptameric context alone was summed for each nucleotide position of each region to estimate the expected substitution frequency across each region of interest. The number of expected substitutions as derived from the heptamer substitution model for a given region was compared to observed substitutions for the European subpopulation within the Phase 1 of the 1000 Genomes Release. We performed comparisons across UTR pG4 G-tracts, pG4 non-G-tract Gap sequences, and constraint-matched non-pG4 UTRs for all pG4s, and the subset of rG4-seq supported pG4 motifs. Because the model does not adjust for methylation at CpG dinucleotides, all CpG dinucleotide positions within UTRs were removed from consideration. Statistical significance was determined by randomly sampling a set of genomic positions from each region of interest with replacement, matching the original combined size of each region, over 10,000 iterations to produce a distribution of observed/expected ratios for each region of interest. $P$ values for each region (pG4, rG4, and gap sequences) were calculated as the proportion of the observed/expected ratios obtained from the above bootstrapping procedure that were less than a matched set of observed/expected ratios using all 5′ and 3′ UTR genomic positions obtained by the same procedure.

**pG4 isoform expression across tissues in GTEx**. The median expression of each annotated RNA transcript (as measured in units of TPMs) in each tissue context was downloaded from GTEx v7. Median TPMs for each transcript were extracted for all pG4- or non-pG4 containing transcript for each pG4 gene, the highest expressed pG4 or non-pG4 isoform was selected, and a threshold of 1 TPM was used to determine expression within a specific tissue context. pG4 transcripts were deemed constitutive if only one of the pG4, or none of the non-pG4 transcripts exceeded this threshold, and labeled alternative if both the pG4, and non-pG4 transcripts exceeded this threshold. Significance of the distribution of alternative vs. constitutive UTR pG4-encoding genes was assessed by randomly assigning pG4 and non-pG4 transcripts each gene, maintaining the number of transcript isoforms encoded by each gene constant with the condition that each gene should contain at least 1 pG4-encoding transcript. The distribution of the ratio of alternative to constitutive pG4 genes from the randomly distributed pG4 transcripts was then computed over 10,000 iterations to obtain a $P$ value for the true ratio of alternative to constitutive UTR pG4-encoding genes.

**cis-eQTL and protein-binding enrichment**. Significant variant-gene pairs were obtained from GTEx release version 7 (URL: https://gtexportal.org/home/datasets) constituting the set of nominally significant cis-eQTLs. Lead cis-eQTL variants for each gene were defined as the variant with the lowest $P$ value for each gene, from the set of all significant variants in each tissue context separately. The set of lead and nominally significant variants was overlapped with UTR pG4 and non-pG4 regions of the UTR, and the number of significant cis-eQTL variants per region was compared to the number of non-significant tested SNPs occupying the same region to determine the proportion of cis-eQTLs compared to non-cis-eQTL SNPs. UTRs with cis-eQTLs not associated with changing the expression of the parent gene were excluded this analysis. Enrichment of cis-eQTLs was computed using a two-sided

Fisher exact test. The set of causal eQTL candidates were obtained directly from the Supplemental material of Brown et al.[30], and enrichment statistics were computed using GTEx v6p tested SNPs instead of v7 to match the data used in that study.

The direction bias of nominally significant cis-eQTLs within UTR pG4 G-tracts vs. non-G-tract variants was computed by binarizing the normalized effect size precomputed for each QTL by GTEx, and comparing the proportions of QTLs in each feature with either a positive effect, or negative effect on gene expression for each possible cis-eQTL annotation across all tissue contexts combined. Statistical significance was determined by a two-sided Fisher exact test.

High-confidence protein-binding sites were obtained from ENCODE CLIP-seq summaries and only peaks called with an irreproducible discovery rate = 1000 were used for downstream enrichment analyses as determined by ENCODE[32]. Overlapping binding sites for multiple proteins were collapsed into a single protein-binding site, and the density of unique binding sites overlapping UTR pG4 regions compared to non-pG4 regions of the UTR was compared by dividing the number of CLIP-seq peaks overlapping each feature by the total number of nucleotides in each region. Significance was assessed using a chi-square test with 2 degrees of freedom.

Proteins whose binding sites are enriched for pG4 overlaps were computed using a hypergeometric test, by comparing the proportion of set of pG4 containing vs. non-pG4 binding sites for a given RBP compared against the background proportion of all UTR CLIP-seq peaks containing a pG4 sequence. The significance of pairwise overlaps between protein-gene targets was also computed using a hypergeometric test to assess the degree that one protein's pG4-binding genes were also targets for another protein.

**Gene expression with RBP knockdown in ENCODE**. Processed differential gene expression tables for K562 and HepG2 were obtained directly from ENCODE (https://www.encodeproject.org/) for each of the pG4-enriched binding proteins and their respective knockdown experiments. For each experiment, a gene was considered differentially expressed at an FDR threshold of <0.05. Genes from ENCODE differential expression tables were annotated as either a pG4 gene or non-pG4 gene on the basis of whether they encoded for a transcript isoform possessing a UTR pG4 sequence in either the 5′ UTR or 3′ UTR. The odds ratio for being significantly differentially expressed was calculated by comparing the ratio of pG4 to non-pG4 genes reaching statistical significance for differential expression between shRNA knockdown of the RBP, and the control for each protein separately. Statistical significance was determined by Fisher's exact test and FDR was controlled at 0.001 by applying the Benjimini–Hochberg procedure to the resultant $P$ values for each cell line. The direction of effect on pG4 gene expression for protein knockdown to cause an increase or decrease in pG4 gene expression was determined by taking the median value for log2-fold change in expression for all pG4-containing genes measured in a given experiment.

**Variants in ClinVar and the NIH-GWAS Catalog**. The April 2019 release of ClinVar was obtained from ftp://ftp.ncbi.nlm.nih.gov/pub/clinvar/. Using these variant annotations, we identified a subset of disease-associated genes as any gene with at least one variant having a Pathogenic or Likely Pathogenic annotation. These genes were used to subset the ClinVar database and all variants spanning 40 nucleotides or less were overlapped with UTR regions to assess for enrichment in pG4 sequences. Insertions or deletions spanning greater than 40 nucleotides were not considered in this analysis, nor were any variants with an annotation of Benign or likely benign in Clinvar. The number of variants across each region was then divided by the total number of bases in each respective region to estimate of the density of variation in a given region. The odds ratios for the number of single nucleotide variants compared to the number of bases in a given region were then compared using a two-sided Fisher exact test.

For identification of GWAS-implicated SNPs affecting annotated UTR pG4 sequences, publicly available phenotype-associated SNPs from were obtained from the NIH-EBI GWAS Catalog. Genomic coordinates for GWAS SNPs were converted from hg38 to hg19 coordinates using the NCBI Genome Remapping Service (https://www.ncbi.nlm.nih.gov/genome/tools/remap). This set of lead GWAS SNPs was used to identify nearby linked SNPs in high LD using the Linkage Disequilibrium Calculator tool from the Ensembl GRCh37 website using a 50KB window surrounding each lead GWAS SNP and selecting the set of SNPs with $r^2 >$ 0.85 using the GBR population of the 1000 Genomes Project.

**Allele-specific expression for GWAS variants**. RNA-seq libraries were trimmed using TrimGalore[53]. Reads were aligned to the GRCh37 human genome using STAR (version 2.7.0c) with the WASP-filtering option, and matched whole-genome sequencing variant files obtained from GTEx for skeletal muscle, thyroid, fibroblast, esophagus, and tibial nerve tissue samples. Reads that did not pass WASP-filtering were removed from the resulting aligned bam files. Polymerase chain reaction duplicates were removed using the python script remove_duplicates.py included in the WASP version 0.3.3 pipeline (https://github.com/bmvdgeijn/WASP). Read counts matching the reference and alternate alleles in the resultant WASP-filtered bam files were compiled using bcftools mpileup across UTR pG4 variants. A beta-binomial model was fitted using the R VGAM package[54] for each variant across all heterozygous samples identified using matched whole-genome

sequencing from GTEx to estimate the ratio of reference reads to alternate reads. Estimates of statistical significance were obtained by using a likelihood ratio test comparing the log-likelihood of the observed count distribution for each variant using the beta-binomial estimate for $\rho$ vs. the null hypothesis of no bias ($\rho = 0.5$).

**Statistics**. Data were analyzed and statistics performed using $R$ (version 3.5.0) and $Python$ (version 3.7 and 3.6.1). Significant differences are noted by asterisks (***).

**Ethics statement**. Our analysis uses publicly accessible, de-identified data sources.

**Reporting summary**. Further information on research design is available in the Nature Research Reporting Summary linked to this article.

## Data availability

Preprocessed data, and instructions for how to access public data resources used in this study that can be used to regenerate the primary figures of this analysis have been uploaded to https://bitbucket.org/biociphers/g4-paper-2019/src/master/. A subset of the processed publicly available data underlying Figs. 3–5 are included in this repository, with associated instructions on how to access other data as required to regenerate these figures where necessary. This repository also contains a link to a Source Data file which contains raw data underlying Figs. 1b–d, 3a, b, 4a–d, 5a, c, d, and Supplementary Figs. 5 and 7. Genetic variation data from The Genome Aggregation Database version 2 release are available from https://gnomad.broadinstitute.org/downloads. Gene expression and cis-eQTL mapping data from the Genotype Tissue Expression Project version 7 release are available from the GTEx Portal website https://gtexportal.org/home/. RNA-seq data used for allelic imbalance analysis are available from dbGaP (phs000424.v7.p2). RNA–protein binding interaction data can be retrieved from the ENCODE Consortium https://www.encodeproject.org/. GWAS-associated variation data can be accessed from the NIH-EBI GWAS Catalog https://www.ebi.ac.uk/gwas/. A copy of the filtered ClinVar database used to generate Fig. 5a is included in the code repository from April 2019. The most updated version of disease-associated genetic variant annotations are also available from ClinVar https://www.ncbi.nlm.nih.gov/clinvar/.

## Code availability

All analysis scripts used to generate the primary results and figures reported in this study are publicly available from https://bitbucket.org/biociphers/g4-paper-2019/src/master/.

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

## Acknowledgements

We thank Dr. Benjamin Voight, Dr. Casey Brown, Matthew Gazzarra, and Joseph Aicher for insightful discussions and thoughtful feedback. D.L. is supported by the NIH grant 5T32HG000046-20. Y.B. and D.L. work was supported by R01 GM128096.

## Author contributions

D.L., L.G., and Y.B. conceived and designed the project. D.L. performed the analyses under the guidance of L.G. and Y.B. D.L. wrote the paper, and all authors contributed to editing the paper.

## Competing interests

The authors declare no competing interests.
