## [Peer Review File · Nature Communications]

Reviewers' comments:

Reviewer #1 (Remarks to the Author):

David Lee and colleagues characterize the constraints and enrichment of functional association in proximity of putative G-quadruplexes, identifying these sequences to be under negative selection and enriched for cis-eQTLs and RNA-binding protein interaction sites. For the study, pG4s are predicted using the canonical G4 motif and further filtered using rG4-seq, thus the authors are only considering the most canonical G4s. The study convincingly shows the biological significance of canonical G4s in UTRs and provides insight into possible regulation of G4 functionality. It will be important in future studies to consider G4s more widely, which will likely lead to more protein binding partners and slightly differing characteristics. However this study is of great importance, particularly given recent discussions within the community as to whether G4s fold in vivo, as it provides strong evidence for the biological importance of these sequences.

Some modifications should be made to clarify and ensure reproducibility of the study:

- 'A gene was considered to harbor a 'constitutive' pG4 sequence when all annotated protein-coding transcript isoforms contained at least one pG4.' According to this definition, two isoforms of the same gene could have different pG4s and would still be considered to have a constitutive pG4, even though the pG4s might have different functional roles and recruit different proteins. Do most genes with such constitutive pG4s have the same pG4 for all isoforms? Are there more or less identical and non-identical constitutive pG4s than expected by chance? This information should be added to the manuscript. When constitutive pG4s do not have the same pG4 in all isoforms, do the pG4s look similar (eg loop lengths and sequences)? The section title is a bit too strong as compared to the results presented: it is not because a pG4 is alternative in a gene, that it is isoform-specific.

- Reproducibility: more details are necessary to ensure reproducibility of the study. Were all transcripts considered, or only transcripts of a certain TSL? For the identification of G4 sequences used for the rest of the analysis, the authors should provide one or more Shell scripts, Makefiles or anything else to represent the computational pipeline in a formalized way including the in-house scripts. This could be added as supplementary material or on a platform like Zenodo. Alternatively, the authors could provide the list of all pG4s and their coordinates.

- More references are necessary in some sections of the text (for example in the beginning of Section 'pG4s in 3'UTRs of disease-associated genes are enriched for variation')

Minor comments:

- Ensemble -> Ensembl

- Page 14: turn operate?

- On page 8, section 'pG4 motifs in the 5' and 3'UTR are enriched for cis-eQTLs': statistics on enrichment would be easier to digest in a Table (same thing for Figure 3).

- In Figure 1d, G4 G1, G4 G2, G4 G3 could be a bit confusing since G4 and G1, G2, G3 do not refer to the same concept. Their meaning should be spelled out.

Reviewer #2 (Remarks to the Author):

Lee et al. present an analysis of G-quadruplex sequences within UTRs. This work makes a nice addition to the growing evidence that variants in 'near-coding' regions are important for human disease. Although this analysis is thorough, I have a number of important concerns and suggestions that need to be addressed.

In addition to the specific points raised below, I have concerns over the power of the presented

analyses. This is not the fault of the authors who have tried admirably to incorporate available data, but is a consequence of UTR regions being small and variation within them hence rare. This means that it is difficult to draw conclusions from the constraint analysis as the confidence intervals for each variant set are very large, and the final disease analysis in the paper is quite weak, with only a single example variant discussed.

I also have concerns over the reproducibility of the analysis in the presented form. The methods section is very patchy, there are repeated mentions of custom scripts and there are no sections detailing data or code availability.

Major concerns:

[1] Throughout the first section of the results, the authors use a set of 'constraint matched' UTRs. The methods suggest that they use the raw obs/exp values from gnomAD.

a. The first mention of this does not reference the recent gnomAD pre-print.

b. Instead of using obs/exp, the authors should be using the upper bound of this value, or the LOEUF score, as this corrects for gene length and hence is a more robust measure of LoF constraint.

c. I do not understand what is being shown in Supplementary Figure 1. What is being plotted should be explained further either in the legend or through more informative axis labels.

d. Can the authors justify why they are using these matched UTRs rather than non-pG4 sequences in the pG4 containing transcripts as a control?

[2] For the variant frequency analysis in Figure 1b it is unclear whether this is looking at the number of variant sites or the allele frequency/allele counts of observed variants. This should be clarified in the main text and the methods.

[3] I have major concerns about the constraint analysis presented in Figure 1c. My main concern is that the authors do not account for base-level methylation in the underlying model. I seriously doubt that 5'UTRs have less than half of the expected number of variants when compared to intergenic regions. The constraint here is inflated given that 5'UTRs are known to be hypomethylated compared to other genomic regions. This means that C>T variants at CpG sites will occur *far* less frequently than in other regions (and far less frequently than expected in the model). This could also affect the comparisons made between G-tracts and G-gaps. The authors should refine this model to take into account base-level methylation at CpG sites or remove these from the analysis.

[4] The methods for the MAPS scores calculated are very sparse with no details of how the underlying model was calculated.

a. Is this based on the same heptamer model as the constraint analysis? If so, it should also be adjusted to also account for base-level methylation.

b. Either way, the methods describing this should be expanded.

c. Is this calculated using 1000 genomes or gnomAD?

[5] The MAPS analysis should include additional control sites that are G triplets that are not part of pG4 sequences. This would reassure that none of the results are a consequence of different underlying sequence contexts.

[6] The statement in the abstract (and discussion) referring to the strength of selection on UTR pG4s should be clarified or removed. The sentence currently suggests that the selection against all UTR pG4s is equivalent to missense variants. In reality, this claim is limited to the central G and upstream of Clinvar disease genes. In addition, all of the MAPS results have very large CIs that in the most part bridge the entire spectrum between synonymous and missense variants.

[7] The authors do not assess how different transcripts affect the constraint and MAPS analyses. If

pG4 sequences that only fall on transcripts that are not highly expressed were removed, would this increase the constraint/selections signal of these elements? Or if only "constitutive" pG4s were considered?

[8] I have various concerns relating to the eQTL analyses

a. In the methods the authors state that "significant variant-gene pairs were obtained from GTEx" – details should be added as to what constitutes a 'significant' association. Was a large multiple testing burden corrected for?

b. The authors should also detail what is included in the 'nominal' set of variants – are these all variants at a locus. Was this set also pruned to remove linked SNPs at a single locus?

c. Similarly, for the analysis presented in Fig 3b – does this only include a single SNP per locus?

[9] Regarding the ClinVar analysis. Did the authors make any attempt to filter the analysed variants? I anticipate that any true enrichment will be strengthened if only high-quality ClinVar variants are taken into account. For example, ClinVar contains many erroneous 'literature only' curations with no evidence for actually being pathogenic. The authors should filter variants to only those with associated evidence for pathogenicity. This filtered set should be used for both the gene selection and the enrichment analysis within these genes. Details of all ClinVar variants, including whether they are in the rG4 and RBP pG4 sets, should also be included in a supplementary table.

[10] Secondly, with relation to the ClinVar analysis, did the authors consider any distance correction? Given that the majority of clinical sequencing to date has focused on coding regions (as the authors note) the most likely captured regions of the UTR (and hence most likely to be represented in ClinVar) are those closest to the coding sequence. The authors should either test enrichment against distance matched UTR sequences or show that there is no bias in either the ClinVar data or the positioning of the G4 sequences in each set.

[11] Finally, on the ClinVar analysis – I had gathered that the rG4 set was a sub-set of the wider pG4 set of sequences. Why then are their fewer ClinVar variants included in the 'All pG4' than 'rG4-seq G4' sets in figure 5a?

Minor comments:

[1] The main text of the manuscript is very long with a lot of repetition. I think it could be condensed significantly whilst still conveying the key messages. In particular, the discussion is far too long: this should not repeat every single analysis and finding (and should not include references to figures), rather it should draw some overall conclusions.

[2] Half of the introduction to the manuscript describes the full methods and results of the paper. This much detail is not needed in the introduction. I would recommend cutting this down to one succinct paragraph of the key analyses and conclusions.

[3] The introduction should include a section to detail what is currently known about noncoding variation in human disease.

[5] In the first section of the methods, when the authors detail how many genes contain pG4 and rG4 sequences, they should include the number of genes/transcripts that were analysed and the overlap between 5' and 3' pG4s (i.e. how many genes contain both vs none?).

[6] I am unclear as to why the authors use rG4 as the shorthand way to refer to experimentally validated sequences. Wouldn't vG4 or eG4 be more appropriate?

[7] "We hypothesised that allele frequencies...should be skewed towards more rare variation" – this is not a new approach and should be appropriately referenced.

[8] The description of pG4 vs rG4 in figure 1 should be described earlier in the legend as it is essential to understand 1b.

[9] For the counts of "alternative" pG4 genes, the total number of genes considered should be included and the overlap between the 5' and 3' UTR counts should be included (i.e. how many genes are "alternative" for other 5' and 3' pG4s?).

[10] A control point should be added to each row of Fig 2b to show the fraction of all transcripts that are expressed in each tissue. For example, most transcripts show higher expression in testes than other tissues. As a related point, I am not in any way an expert on RNAseq data, but should the expression level be normalised per tissue rather than taking a cutoff of 1 TPM for all tissues?

[11] The authors should include a supplementary table showing all GWAS catalogue variants within pG4s.

Reviewer #3 (Remarks to the Author):

This manuscript presents in-depth bioinformatic analyses of G-quadruplex (G4) in untranslated regions. The authors reported evidence of negative selection on UTR putative G4s. They also observed that G4s are often located in alternative isoforms and overlap cis-eQTLs. Using protein-binding data, the authors identified RNA-binding proteins that may bind to pG4s. In addition, analysis of GWAS variants revealed an enrichment for disease-associated variants in 3' UTR G4s. These results suggest that G4s in 3' UTRs are likely functional elements in gene regulation. Overall, the potential function of G4s in 3' UTRs is a very interesting question. The analyses in this study were carefully executed and well described. The authors presented compelling bioinformatic evidence to support that G4s in 3' UTRs may be functional. I have the following comments to improve the manuscript.

The finding that G4-containing transcripts were often isoform-specific is interesting. However, there is much room to strengthen and extend this observation. Specifically, what fraction of pG4s is located in the longer 3' UTR regions? What are the functional relevance? Could pG4s affect alternative polyadenylation? Given their enrichment relative to cis-eQTLs, could 3' UTRs pG4s affect RNA expression through regulating RNA turnover? Given the protein binding data analyzed in this study, the above questions could be examined in conjunction with a few putative G4-binding proteins.

In analyzing protein-binding data, the authors may want to pinpoint the cross-linking sites in the CLIP reads, and examine the relative enrichment of pG4s around cross-linking sites (which provide stronger evidence of direct protein-RNA interaction).

For many proteins in ENCODE, RNAseq data following their knockdown in cells are also available. The authors should take advantages of these data and further investigate interesting G4-related proteins and the potential functional consequence of such relationships.

The finding for HSPB7 is interesting. Can the authors expand the analysis to other genes and examine whether similar observations hold for more genes?

Detailed response to reviewers' comments

Manuscript: *Integrative analysis reveals RNA G-Quadruplexes in UTRs are selectively constrained and enriched for functional associations*

Reviewer #1 (Remarks to the Author):

David Lee and colleagues characterize the constraints and enrichment of functional association in proximity of putative G-quadruplexes, identifying these sequences to be under negative selection and enriched for cis-eQTLs and RNA-binding protein interaction sites. For the study, pG4s are predicted using the canonical G4 motif and further filtered using rG4-seq, thus the authors are only considering the most canonical G4s. The study convincingly shows the biological significance of canonical G4s in UTRs and provides insight into possible regulation of G4 functionality. It will be important in future studies to consider G4s more widely, which will likely lead to more protein binding partners and slightly differing characteristics. However this study is of great importance, particularly given recent discussions within the community as to whether G4s fold in vivo, as it provides strong evidence for the biological importance of these sequences.

Some modifications should be made to clarify and ensure reproducibility of the study:

-‘A gene was considered to harbor a ‘constitutive’ pG4 sequence when all annotated protein-coding transcript isoforms contained at least one pG4.’ According to this definition, two isoforms of the same gene could have different pG4s and would still be considered to have a constitutive pG4, even though the pG4s might have different functional roles and recruit different proteins. Do most genes with such constitutive pG4s have the same pG4 for all isoforms? Are there more or less identical and non-identical constitutive pG4s than expected by chance? This information should be added to the manuscript.

When constitutive pG4s do not have the same pG4 in all isoforms, do the pG4s look similar (eg loop lengths and sequences)? The section title is a bit too strong as compared to the results presented: it is not because a pG4 is alternative in a gene, that it is isoform-specific.

We thank the reviewer for the thoughtful and constructive feedback. Our point-by-point responses to questions and concerns follow below.

In response to the reviewer’s concern regarding the section title corresponding to isoform-restricted pG4 motifs, we have changed the title to better represent our results:

“Most pG4 motifs within untranslated regions of mRNA are isoform-restricted”

We have checked our set of “constitutive” pG4-containing isoforms. For both the 5’ UTR and 3’UTR the majority of constitutive pG4 genes produce transcripts that share the same pG4. Out of 620 “constitutive” 5’UTR pG4 genes we find 36 genes produce transcripts with nonidentical pG4s (that differ either in the number or positions of pG4 sequences). For the 3’UTR this fraction is 75 / 1275. When constitutive pG4s do not share the same pG4 across all isoforms, 13 out of the 36 5’UTR pG4 differ by the addition / subtraction of pG4 motifs from the annotated UTR sequence. 20 out of the 75 3’UTR pG4 differ by the addition / subtraction of pG4 motifs from the annotated UTR sequence. We have added this additional information to the main text, however as these are a minority of the pG4s identified and mapped by our study, we did not pursue this inquiry further.

-Reproducibility: more details are necessary to ensure reproducibility of the study. Were all transcripts considered, or only transcripts of a certain TSL? For the identification of G4 sequences used for the rest of the analysis, the authors should provide one or more Shell scripts, Makefiles or anything else to represent the computational pipeline in a formalized way including the in-house scripts. This could be added as supplementary material or on a platform like Zenodo. Alternatively, the authors could provide the list of all pG4s and their coordinates.

In this study we considered all transcripts annotated in Ensembl release v. 75 with the “transcript_biotype” annotation of “protein-coding” without further filtering based on TSL. We chose to maximize the set of annotated transcripts considered in this study to increase the set of canonical pG4 sequences that could be identified by our text-matching approach.

We thank the reviewer for the reviewer for the additional comments regarding the reproducibility of our study. We agree and have amended the methods section to clarify our approach, and have uploaded all scripts / RData objects used to produce the figures in our study for public release to ensure reproducibility and transparency to the following bitbucket repository: <https://bitbucket.org/biociphers/g4-paper-2019/src/master/>

-More references are necessary in some sections of the text (for example in the beginning of Section ‘pG4s in 3’UTRs of disease-associated genes are enriched for variation’)

We agree and have revised the text and added in additional citations throughout as necessary.

Minor comments:

-Ensembl -> Ensembl

-Page 14: turn operate?

- On page 8, section ‘pG4 motifs in the 5’ and 3’UTR are enriched for cis-eQTLs’: statistics on enrichment would be easier to digest in a Table (same thing for Figure 3).

- In Figure 1d, G4 G1, G4 G2, G4 G3 could be a bit confusing since G4 and G1, G2, G3 do not refer to the same concept. Their meaning should be spelled out.

We thank the reviewer for these careful observations and agree with these suggested changes. We have added a supplementary table containing enrichment statistics for Figure 3 (**Suppl. Table 3**), and amended labeling in Fig 1d. Additionally we have added text to the Figure legend to clarify this point.

Reviewer #2 (Remarks to the Author):

Lee et al. present an analysis of G-quadruplex sequences within UTRs. This work makes a nice addition to the growing evidence that variants in 'near-coding' regions are important for human disease. Although this analysis is thorough, I have a number of important concerns and suggestions that need to be addressed.

In addition to the specific points raised below, I have concerns over the power of the presented analyses. This is not the fault of the authors who have tried admirably to incorporate available data, but is a consequence of UTR regions being small and variation within them hence rare. This means that it is difficult to draw conclusions from the constraint analysis as the confidence intervals for each variant set are very large, and the final disease analysis in the paper is quite weak, with only a single example variant discussed.

We thank the reviewer for their thorough reading of our manuscript. We have worked to address each of the reviewer's concerns and believe that these contributions have strengthened the presented analyses. While the numbers of UTR pG4 are limited, we were able to find evidence for heightened selective constraint by three metrics: 1) a reduction in allele frequencies, 2) a decrease in the number of polymorphic sites, and 3) an enrichment for rare variation (MAPS) that, taken together, support our conclusion that UTR pG4 sequences are under heightened selective constraints giving us confidence in our analysis. Additionally we have expanded our initial analysis of disease-relevance to present additional disease-associated variants implicated by GWAS by including SNPs that are in high-LD with GWAS-linked SNPs in our analysis of allelic imbalance. Finally we would also point to another recent preprint posted on BioArXiv from the MacArthur group using a similar approach and the same data to assess selective constraint against 5'UTR uORFs¹.

I also have concerns over the reproducibility of the analysis in the presented form. The methods section is very patchy, there are repeated mentions of custom scripts and there are no sections detailing data or code availability.

We agree with the reviewer that our methods and code availability can be improved considerably as this point was also raised by Reviewer #1. We substantially added to our methods section, and have released all analysis scripts and RData objects used to generate each of the primary figures in the manuscript. We hope that the reviewer finds this adequately addresses their concerns regarding the reproducibility of our analysis.

Major concerns:

[1] Throughout the first section of the results, the authors use a set of 'constraint matched' UTRs. The methods suggest that they use the raw obs/exp values from gnomAD.

- a. The first mention of this does not reference the recent gnomAD pre-print.
- b. Instead of using obs/exp, the authors should be using the upper bound of this value, or the LOEUF score, as this corrects for gene length and hence is a more robust measure of LoF constraint.
- c. I do not understand what is being shown in Supplementary Figure 1. What is being plotted should be explained further either in the legend or through more informative axis labels.
- d. Can the authors justify why they are using these matched UTRs rather than non-pG4 sequences in the pG4 containing transcripts as a control?

We thank the reviewer for highlighting the ambiguities of this section as it was written, and we agree that using the upper bound LOEUF score is more appropriate for our analyses. We have included a citation for the gnomAD pre-print with the first mention of the obs/exp values. We have repeated all of our analyses previously relying on the raw obs/exp statistic now using the upper bound of the observed / expected metric (LOEUF) as suggested by the reviewer, and amended the main text and methods section to reflect these updates. Specifically this repeated analysis has nominally changed our results for **Fig. 1b**, **Fig 1c**, **Suppl Fig. 1** and **Suppl Fig. 2**. We have also expanded the figure caption of Supplementary Figure 1 to clarify what is being shown.

Regarding why we are using matched UTRs rather than non-pG4 sequences within pG4-containing transcripts, we intended to maximize the number of non-pG4 variants that can be used for comparison in our analysis. Drawing clear conclusions from a comparison of non-pG4 G-tracts within pG4-containing transcripts would be more difficult, since it is possible that nearby non-pG4 G-tracts may also participate in longer-loop G4 formation, or facilitate the formation of multiple overlapping G4 structures that rely on different combinations of nearby G-tracts as has been previously reported^{2,3}.

[2] For the variant frequency analysis in Figure 1b it is unclear whether this is looking at the number of variant sites or the allele frequency/allele counts of observed variants. This should be clarified in the main text and the methods.

We thank the reviewer for pointing out this ambiguity in the manuscript as it was written. In this analysis we use allele frequencies, and have amended the text and methods within the manuscript to clarify this point.

[3] I have major concerns about the constraint analysis presented in Figure 1c. My main concern is that the authors do not account for base-level methylation in the underlying model. I seriously doubt that 5'UTRs have less than half of the expected number of variants when compared to intergenic regions. The constraint here is inflated given that 5'UTRs are known to

be hypomethylated compared to other genomic regions. This means that C>T variants at CpG sites will occur *far* less frequently than in other regions (and far less frequently than expected in the model). This could also affect the comparisons made between G-tracts and G-gaps. The authors should refine this model to take into account base-level methylation at CpG sites or remove these from the analysis.

We thank the reviewer for this excellent point. The number and frequency of variants within CpG sites will occur far less frequently than in other regions and this was observed in our data as well. We have repeated the experiment removing all CpG dinucleotides from 5' and 3' UTR sequences from consideration in the constraint analysis, and amended Figure **1b**, **1c**, and **1d** accordingly. We find that the heptamer model fit of UTRs to the intergenic noncoding model is much improved after controlling for CpG sites for both the 5'UTR and 3'UTR (~90% of expected sites are observed in 5'UTR, ~85% of expected sites are observed in 3'UTR). We further note that, as the reviewer suspected, the confounding influence of CpG sites was significantly affecting the comparisons made between G-tracts and gap sequences. With these regions removed from consideration we find that gap sequences are not significantly different from 5' / 3' UTR in general. We have changed the main body of the text and our methods to reflect these revisions.

[4] The methods for the MAPS scores calculated are very sparse with no details of how the underlying model was calculated.

- a. Is this based on the same heptamer model as the constraint analysis? If so, it should also be adjusted to also account for base-level methylation.
- b. Either way, the methods describing this should be expanded.
- c. Is this calculated using 1000 genomes or gnomAD?

We agree with the reviewer's assessment that our original description of the MAPS model was inadequately detailed. We note in methods that we are using a three-mer model for the MAPS analysis. We have significantly expanded our methods section describing how the model was developed, and have clarified in the methods and main text that the model was applied to gnomAD data. As in the concern raised above regarding differences in base-level methylation, we have similarly removed CpG sites from consideration from this analysis and updated the figure accordingly.

[5] The MAPS analysis should include additional control sites that are G triplets that are not part of pG4 sequences. This would reassure that none of the results are a consequence of different underlying sequence contexts.

We thank the reviewer for this good suggestion. We have included additional control sites that are G/C-triplets that are not part of the pG4 sequences to control for the possible effect of different underlying sequence contexts. We note that this analysis showed that for the 1st, 2nd, and 3rd positions within each trinucleotide G/C tract, we do not observe significant deviation from synonymous variation by MAPS score (permuted $P=0.0171$). Thus, the enrichment for

singletons is specific to UTR G-tracts involved in the larger pG4 motif. We have included this analysis in our manuscript with the addition of **Suppl. Fig. 3** in our revised manuscript.

[6] The statement in the abstract (and discussion) referring to the strength of selection on UTR pG4s should be clarified or removed. The sentence currently suggests that the selection against all UTR pG4s is equivalent to missense variants. In reality, this claim is limited to the central G and upstream of Clinvar disease genes. In addition, all of the MAPS results have very large CIs that in the most part bridge the entire spectrum between synonymous and missense variants.

We agree with the reviewer that our statements in the abstract and discussion referring to the strength of selection on UTR pG4s is too strong. We have qualified the statement in our abstract and discussions to more accurately reflect our analysis.

[7] The authors do not assess how different transcripts affect the constraint and MAPS analyses. If pG4 sequences that only fall on transcripts that are not highly expressed were removed, would this increase the constraint/selections signal of these elements? Or if only “constitutive” pG4s were considered?

We thank the reviewer for this suggestion. To address the possibly different constraints on different transcript sets, we have evaluated the relative degree of constraint (as measured by MAPS) on constitutive vs. alternative pG4 UTR regions. Intriguingly, this analysis showed that selective pressures the central G-tract G in alternative pG4 sequences tend to be increased compared to constitutive pG4 sequences, implying that these alternative pG4 sequences tend to be more functionally important. Unfortunately there were not enough variants in constitutive pG4 G-tracts for our analysis of constraint acting over these sequences to reach statistical significance. Given the heightened constraint acting over alternative pG4 G-tract central positions, we have amended **Fig 1d** to include this data, and moved the original assessment of constraint of pG4 sequences in the top 1/3 of protein-coding genes to **Suppl. Fig. 3**.

[8] I have various concerns relating to the eQTL analyses

- a. In the methods the authors state that “significant variant-gene pairs were obtained from GTEx” – details should be added as to what constitutes a ‘significant’ association. Was a large multiple testing burden corrected for?
- b. The authors should also detail what is included in the ‘nominal’ set of variants – are these all variants at a locus. Was this set also pruned to remove linked SNPs at a single locus?
- c. Similarly, for the analysis presented in Fig 3b – does this only include a single SNP per locus?

We used the set of “significant variant-gene associations” provided in the GTEx v7 release. According to GTEx, this set of significant variant-gene associations are determined to be significant for each gene by a permutation-adjusted P-value, as calculated by GTEx with FDR controlled at ≤ 0.05 for each variant-gene association in each tissue context. We take the set

of significant variant-gene pairs in GTEx that cross this FDR threshold and use this as our set of “nominally significant” set of variants.

We note that this set of nominally significant variants were not LD-pruned. We agree with the reviewer that LD-can affect our estimates of enrichment since nearby-linked nominally significant cis-eQTLs may artificially increase our estimate of the ratio of cis-eQTL SNPs vs. non-cis-eQTL SNPs within UTR pG4 regions. However, since the relative size of UTR pG4 motifs (23 - 30 nt long) is compact in comparison to UTRs in general, we do not anticipate that LD-pruning to remove nearby linked variants will significantly change our results that cis-eQTLs are enriched within UTR pG4 sequences. Because more non-pG4 cis-eQTL SNPs are more likely to be nominally significant due to LD compared to pG4 SNPs (due to the difference in feature sizes) we suspect that LD-pruning our set of nominally significant SNPs will even further increase our estimate of cis-eQTL enrichment in pG4 regions since we anticipate that more nominally significant SNPs will be removed from the non-pG4 regions.

Nevertheless, we find that 9 cis-eQTLs in our analysis are located within the same UTR pG4 sequence for the 5'UTR pG4, and 3 variants are located within the same UTR pG4 sequence for 3'UTR pG4. When we remove these variants from our analysis, the results continue to demonstrate significant enrichment of cis-eQTLs within UTR pG4, despite only accounting for potentially linked SNPs across UTR pG4 cis-eQTLs and not removing linked SNPs from non-pG4 regions. Given the above, we concluded that the analysis does not need to be amended, however we acknowledge the possibility of linked SNPs changing our estimate of cis-eQTL enrichment in UTR pG4 in our discussion.

Finally to further address this possible source of confounding, we have assessed enrichment for a set of causal independent cis-eQTL variants mapping to UTRs as identified by Brown et. al⁴. Here we see that even when only high-confidence independent cis-eQTLs are considered, we continue to observe enrichment in UTR pG4 sequences. We have included this additional analysis in **Fig. 3a** and updated our discussion accordingly.

[9] Regarding the ClinVar analysis. Did the authors make any attempt to filter the analysed variants? I anticipate that any true enrichment will be strengthened if only high-quality ClinVar variants are taken into account. For example, ClinVar contains many erroneous ‘literature only’ curations with no evidence for actually being pathogenic. The authors should filter variants to only those with associated evidence for pathogenicity. This filtered set should be used for both the gene selection and the enrichment analysis within these genes. Details of all ClinVar variants, including whether they are in the rG4 and RBP pG4 sets, should also be included in a supplementary table.

To clarify our analysis, we had initially performed an analysis using only annotated high-quality pathogenic variations within UTRs. However, due to the severe depletion in annotated high-quality pathogenic variants located within non-coding regions of the genome, and within UTRs in particular, we were underpowered to discover any robust associations. We therefore

asked whether we find evidence for enrichment of “VUS”, “Conflicting Interpretations of Pathogenicity”, “Likely Pathogenic”, or “Pathogenic” variants within UTR pG4 of genes hypothesizing that these variants should be enriched within the set of disease-associated genes in ClinVar. We have amended the text to clarify this point, and included the set of ClinVar variants overlapping UTR pG4 sequences as a supplementary table.

[10] Secondly, with relation to the ClinVar analysis, did the authors consider any distance correction? Given that the majority of clinical sequencing to date has focused on coding regions (as the authors note) the most likely captured regions of the UTR (and hence most likely to be represented in ClinVar) are those closest to the coding sequence. The authors should either test enrichment against distance matched UTR sequences or show that there is no bias in either the ClinVar data or the positioning of the G4 sequences in each set.

The reviewer raises an important point that our analysis of ClinVar enrichment within UTR pG4 sequences may be confounded if UTR pG4 sequence features tend to occupy genomic regions that are close to CDS. To test this, we performed simulation studies to test the hypothesis that the UTR pG4 sequences within pathogenic / likely pathogenic genes used in our experiment are closer to annotated CDS than any randomly selected set of positions within the UTRs of disease-associated genes that lack pG4 sequences. We first determined the genomic distance between each annotated UTR pG4 and the closest CDS annotation using Ensembl 75 (n = 1128 for 5'UTR, 1536 for 3'UTR). We then randomly selected a single position from the set of non-pG4 UTRs, and determined the distribution of these positions relative to the closest annotated CDS (n = 13,489 for 5'UTR, 10,482 for 3'UTR). Our simulations show that UTR pG4 features actually tend to be further from annotated CDS sequences compared to a randomly selected positions in this set of genes; we have revised the manuscript and included a supplemental figure (**Suppl. Fig. 5**) showing this analysis. We conclude from these results that it is unlikely that our analysis of enrichment for ClinVar variants within UTR pG4 is biased by their being positioned closer to CDS.

[11] Finally, on the ClinVar analysis – I had gathered that the rG4 set was a sub-set of the wider pG4 set of sequences. Why then are their fewer ClinVar variants included in the ‘All pG4’ than ‘rG4-seq G4’ sets in figure 5a?

Due to the lower number of variants falling within UTRs, for this analysis we used all rG4-seq mapped G4s, including non-canonical G4-forming sequences (long-loops, G-quartets, and G4-bulges) as reported in the Nature Methods 2016 study to increase our power to detect enrichment⁵. We have clarified the methods and results of the text to emphasize this point.

[1] The main text of the manuscript is very long with a lot of repetition. I think it could be condensed significantly whilst still conveying the key messages. In particular, the discussion is far too long: this should not repeat every single analysis and finding (and should not include references to figures), rather it should draw some overall conclusions.

[2] Half of the introduction to the manuscript describes the full methods and results of the paper. This much detail is not needed in the introduction. I would recommend cutting this down to one succinct paragraph of the key analyses and conclusions.

[3] The introduction should include a section to detail what is currently known about noncoding variation in human disease.

We thank the reviewer for this feedback. We have comprehensively edited the main text of the manuscript to make it more concise. In particular we have shortened the Introduction and Discussion sections considerably. We have also expanded the introduction to briefly introduce key points on what is known about noncoding variation in human disease.

[5] In the first section of the methods, when the authors detail how many genes contain pG4 and rG4 sequences, they should include the number of genes/transcripts that were analysed and the overlap between 5' and 3' pG4s (i.e. how many genes contain both vs none?).

We have updated the first section of the methods to include this information. We find a total of 5235 protein-coding genes harboring a 5' or 3' UTR pG4 sequence. Of these genes, 2967 have a 5'UTR pG4, 2835 have a 3'UTR pG4, with 567 genes having both a 5'UTR and 3'UTR pG4 sequence. For rG4-seq validated sequences, we find 243 genes have a 5'UTR rG4, 803 have a 3'UTR rG4, and 16 genes have both.

[6] I am unclear as to why the authors use rG4 as the shorthand way to refer to experimentally validated sequences. Wouldn't vG4 or eG4 be more appropriate?

We thank the reviewer for voicing this concern. We note that "rG4" derives from the set of "rG4-seq" verified G-quadruplexes from the original Kwok et al. 2015 Nature Methods paper and prefer to keep this nomenclature throughout the manuscript so that the relationship between our set of experimentally validated set of pG4 sequences and the experimental technique used to map these sequences is consistent.^{1,5}

[7] "We hypothesised that allele frequencies...should be skewed towards more rare variation" – this is not a new approach and should be appropriately referenced.

We have included a citation for the following three papers for this hypothesis:

Fay, J. C., Wyckoff, G. J. & Wu, C. I. Positive and negative selection on the human genome. *Genetics* 158, 1227–1234 (2001).

Drake, J. A. et al. Conserved noncoding sequences are selectively constrained and not mutation cold spots. *Nature Genetics* 38, 223–227 (2006).

Chen, K. & Rajewsky, N. Natural selection on human microRNA binding sites inferred from SNP data. *Nat. Genet.* 38, 1452–1456 (2006).

If the reviewer has other citations that they feel may be more appropriate we are happy to accommodate their suggestions.

[8] The description of pG4 vs rG4 in figure 1 should be described earlier in the legend as it is essential to understand 1b.

We have amended the legend of Figure 1 to include the definition of a “rG4-seq” validated pG4 sequence earlier in the text.

[9] For the counts of “alternative” pG4 genes, the total number of genes considered should be included and the overlap between the 5’ and 3’ UTR counts should be included (i.e. how many genes are “alternative” for other 5’ and 3’ pG4s?).

The total number of pG4 containing genes (having a pG4 in either the 5’ or 3’ UTR, or both) is 5235. The total number of alternative pG4 genes of these 5235 is 3395. The overlap between genes that are alternative for 5’UTR pG4 and 3’UTR pG4 is 284 genes. We have amended the results section of our manuscript to include this updated information as suggested by the reviewer, and added this information in our methods section.

[10] A control point should be added to each row of Fig 2b to show the fraction of all transcripts that are expressed in each tissue. For example, most transcripts show higher expression in testes than other tissues. As a related point, I am not in any way an expert on RNAseq data, but should the expression level be normalised per tissue rather than taking a cutoff of 1 TPM for all tissues?

To account for differing degrees of alternative transcript expression across tissues, we have adjusted the proportion of alt. pG4 genes expressing both pG4 and non-pG4 isoforms by the total proportion of alternative pG4 genes expressed in each tissue context. Consequently the x-axis in Fig. 2b now represents (for each tissue):

$$\frac{\textit{Prop. of pG4 genes expressing both pG4 and non-pG4 transcript}}{\textit{Prop. of pG4 genes expressed}}$$

By making this adjustment the proportion of alt. pG4 genes expressing both a pG4 and non-pG4 containing transcript is not dependent on the absolute level of transcript expression in a given tissue context.

Furthermore, we note that the use of TPM (transcripts per million) are appropriate in this context, since we are only assessing the relative expression of pG4 containing transcripts within each tissue context, and not comparing their expression across different tissues. Because TPMs are already a measure of relative expression normalized for sequencing depth and differing transcript lengths, they are an appropriate unit of measurement for transcript expression for this analysis⁶.

[11] The authors should include a supplementary table showing all GWAS catalogue variants within pG4s.

We have updated the manuscript with additional analysis of GWAS catalogue variants and included a supplemental table detailing these variants in the revised manuscript.

Reviewer #3 (Remarks to the Author):

This manuscript presents in-depth bioinformatic analyses of G-quadruplex (G4) in untranslated regions. The authors reported evidence of negative selection on UTR putative G4s. They also observed that G4s are often located in alternative isoforms and overlap cis-eQTLs. Using protein-binding data, the authors identified RNA-binding proteins that may bind to pG4s. In addition, analysis of GWAS variants revealed an enrichment for disease-associated variants in 3' UTR G4s. These results suggest that G4s in 3' UTRs are likely functional elements in gene regulation. Overall, the potential function of G4s in 3' UTRs is a very interesting question. The analyses in this study were carefully executed and well described. The authors presented compelling bioinformatic evidence to support that G4s in 3' UTRs may be functional. I have the following comments to improve the manuscript.

The finding that G4-containing transcripts were often isoform-specific is interesting. However, there is much room to strengthen and extend this observation. Specifically, what fraction of pG4s is located in the longer 3' UTR regions? What are the functional relevance? Could pG4s affect alternative polyadenylation? Given their enrichment relative to cis-eQTLs, could 3' UTRs pG4s affect RNA expression through regulating RNA turnover? Given the protein binding data analyzed in this study, the above questions could be examined in conjunction with a few putative G4-binding proteins.

We thank the reviewer for their positive comments and suggestions. We have performed an additional evaluation to assess the distribution of 5' and 3' UTR pG4 sequences within their respective transcripts. Our results show that 5' and 3' UTR pG4 are nearly distributed uniformly across the length of each UTR, without particular bias for being close or further away from the annotated CDS. This is consistent with previous *in vitro* structure mapping studies performed by Kwok et al. showing that there is a relatively uniform enrichment of structure-forming G4 sequences within the 5' and 3' UTRs⁵. We have included this analysis as a supplemental figure 1 in our revised manuscript.

With regards to the possible functional relevance of 3' UTR pG4, we agree with the reviewer and believe they have identified excellent questions regarding the mechanisms of pG4-protein interactions within UTRs. There have been a handful of examples of specific functional 3' UTR pG4 and it is likely that the functional roles of 3' UTR pG4 are diverse^{2,7-10}. We present compelling evidence here that suggests a broad role for 3' UTR pG4 in regulating gene expression, particularly through mediation of RNA-protein binding interactions, however we feel

that a detailed analysis of these functional relationships, while important and interesting, will ultimately open more questions that are well beyond the scope of the current manuscript.

We note however in responding to the reviewer's 3rd point (below) that we do find evidence of changing gene regulation upon knockdown of certain RBPs that appear to bind 3'UTR pG4 sequences frequently. This coupled with previous findings that 3'UTR G4 structures are enriched near polyadenylation sites implies that it is like a subset of these 3'UTR enriched pG4 binding proteins are involved in regulating polyadenylation⁵.

In analyzing protein-binding data, the authors may want to pinpoint the cross-linking sites in the CLIP reads, and examine the relative enrichment of pG4s around cross-linking sites (which provide stronger evidence of direct protein-RNA interaction).

This is a good suggestion and could provide additional information regarding possible mechanisms of protein-pG4 interactions and different patterns of binding for different pG4-binding proteins. Although pinpointing the cross-linking sites will provide an even higher resolution of RNA-protein interactions and raises an interesting question, we have not yet thoroughly addressed the specific distribution of RNA-protein binding sites within / near UTR pG4 sequences at single-nucleotide resolution. Our primary conclusion is that UTR pG4 regions are enriched for these RNA-protein binding interactions compared to non-pG4 forming regions of UTRs. Given that the CLIP peaks we are using from ENCODE have a median width of 51 nucleotides, we feel that the resolution of using the CLIP-peaks is sufficiently high to support our claim that RBP binding sites are enriched over/in close proximity to UTR pG4 sequences. As such we feel that an analysis of exact binding site interactions and their distributions around UTR pG4, while an interesting question, is beyond the scope of the present manuscript - but raises an interesting avenue for future studies.

For many proteins in ENCODE, RNAseq data following their knockdown in cells are also available. The authors should take advantage of these data and further investigate interesting G4-related proteins and the potential functional consequence of such relationships.

We agree with the reviewer that an analysis of accompanying pG4-binding protein knockdown in ENCODE is an interesting question. We have performed a comprehensive analysis of the available ENCODE RBP knockdown data using the subset of RBPs with evidence for enriched UTR pG4 binding by our analysis. We have identified select RBPs with enrichment for pG4 binding that are much more likely to change the expression of a UTR pG4-including gene compared to a non-UTR pG4-including gene. Moreover we have associated with effects of specific RBPs on the overall expression of UTR pG4 containing genes. We have included the results of this analysis in **Suppl. Fig. 4**.

The finding for HSPB7 is interesting. Can the authors expand the analysis to other genes and examine whether similar observations hold for more genes?

We have expanded our search of common variants associated with phenotypic changes in the NIH-EBI GWAS Catalogue to include the set of SNPs that are in high linkage-disequilibrium ($r^2 > 0.85$ in the GBR population from the 1000 Genomes Project) within a 50KB window surrounding each GWAS tag-SNP overlapping a UTR pG4 region. Through this analysis we uncovered an additional 8000 SNPs with at least one association in the NIH-EBI GWAS Catalogue that overlap at UTR pG4 region. Taking this set of SNPs, we looked across 4 additional tissue contexts in GTEx with matched RNA-seq and WGS for evidence of allelic imbalance. Under the requirement that at least 20 samples needed to be heterozygous for each SNP of interest, we find 4 additional SNPs that are in high linkage disequilibrium with a GWAS tag SNP, affect UTR pG4 sequences, and also exhibit significant allelic imbalance. We have included this additional analysis in our revised manuscript to increase the number of examples associating common variation within UTR pG4 sequences, changes in gene expression, and phenotypic consequences as suggested by the reviewer.

References

1. Whiffin, N. *et al.* Characterising the loss-of-function impact of 5' untranslated region variants in 15,708 individuals. *bioRxiv* 543504 (2019). doi:10.1101/543504
2. Huang, H., Zhang, J., Harvey, S. E., Hu, X. & Cheng, C. RNA G-quadruplex secondary structure promotes alternative splicing via the RNA-binding protein hnRNPF. *Genes Dev.* **31**, 2296–2309 (2017).
3. Agarwala, P., Pandey, S., Ekka, M. K., Chakraborty, D. & Maiti, S. Combinatorial role of two G-quadruplexes in 5' UTR of transforming growth factor β 2 (TGF β 2). *Biochimica et Biophysica Acta (BBA) - General Subjects* 129416 (2019). doi:10.1016/j.bbagen.2019.129416
4. Brown, A. A. *et al.* Predicting causal variants affecting expression by using whole-genome sequencing and RNA-seq from multiple human tissues. *Nat. Genet.* **49**, 1747–1751 (2017).
5. Kwok, C. K., Marsico, G., Sahakyan, A. B., Chambers, V. S. & Balasubramanian, S. rG4-seq reveals widespread formation of G-quadruplex structures in the human transcriptome. *Nat. Methods* **13**, 841–844 (2016).
6. Wagner, G. P., Kin, K. & Lynch, V. J. Measurement of mRNA abundance using RNA-seq

- data: RPKM measure is inconsistent among samples. *Theory Biosci.* **131**, 281–285 (2012).
7. Agarwala, P., Pandey, S., Mapa, K. & Maiti, S. The G-Quadruplex Augments Translation in the 5' Untranslated Region of Transforming Growth Factor β 2. *Biochemistry* **52**, 1528–1538 (2013).
 8. Kumari, S., Bugaut, A., Huppert, J. L. & Balasubramanian, S. An RNA G-quadruplex in the 5' UTR of the NRAS proto-oncogene modulates translation. *Nat. Chem. Biol.* **3**, 218–221 (2007).
 9. Subramanian, M. *et al.* G-quadruplex RNA structure as a signal for neurite mRNA targeting. *EMBO Rep.* **12**, 697–704 (2011).
 10. Beaudoin, J.-D. & Perreault, J.-P. Exploring mRNA 3'-UTR G-quadruplexes: evidence of roles in both alternative polyadenylation and mRNA shortening. *Nucleic Acids Res.* **41**, 5898–5911 (2013).

Reviewers' comments:

Reviewer #1 (Remarks to the Author):

The authors have addressed all my comments in a satisfactory manner, thus clarifying the manuscript.

Reviewer #2 (Remarks to the Author):

I would first like to thank the authors for their detailed responses to my comments and for their efforts to increase the reproducibility of their analysis both through data/code availability and expanding their methods section.

Below are some remaining concerns I have with the manuscript:

[1] The authors need to explicitly outline the limitations of their analysis with regard to the low power and largely overlapping confidence intervals. The authors point to the preprint by Whiffin et al. using a similar approach. The authors of this previous study clearly stated where results were non-significant and outlined the limitations regarding power in their discussion. The authors here should do similarly.

Specifically:

(a) The authors should state power and low variant numbers across their analyses as a fourth major limitation in their discussion section.

(b) The authors spend considerable effort discussing the increased selection against the central G of the triplet compared to the outer Gs. The fact remains that this is not a result shown in the data, as the confidence intervals for all of these analyses are largely overlapping. For the authors to retain this section in the manuscript they should explicitly state that there is no significant difference. They should also be careful with their interpretation of this 'result' in the discussion. In addition, the authors should consider referencing this previous work to highlight that a similar approach has been used previously.

[2] I would like to thank the authors for switching from obs/exp to using the LOEUF metric. The authors should, however, include the name 'LOEUF' in the section in brackets to aid readers in comparing across different studies.

[3] I am still unclear as to what Figure 1b is showing. Please can the authors explicitly write in the legend what is being plotted on the y axis and make sure this is clear in the main text.

[4] The authors talk about assessing "the extent of polymorphism within UTR pG4 sequences". I am not sure that the word 'polymorphism' is the right one here.

[5] I think Figure 1c would benefit from lines showing the constraint against synonymous, missense and pLoF variants as well as the existing intergenic line to allow the reader to interpret the level of constraint again pG4s.

[6] Sup. Fig. 4 needs to be referenced in the main text.

[7] I appreciate the authors efforts to demonstrate that removing linked SNPs does not remove their eQTL association results. This analysis should be included in the paper – either in the methods or supplement for other readers with similar concerns (apologies if I missed where this has been included).

[8] Sup. Table 3 should include numbers for non-pG4 regions as well as pG4s.

[9] In the ClinVar enrichment section the sentence "Instead, we hypothesized that variation, in general, is enriched within the UTR pG4 sequences of disease-associated genes" does not appear to make sense. Surely we would expect variation 'in general' to be depleted but 'disease-associated' variation to be enriched?

Reviewer #3 (Remarks to the Author):

none

Reviewers' comments:

Reviewer #2 (Remarks to the Author):

I would first like to thank the authors for their detailed responses to my comments and for their efforts to increase the reproducibility of their analysis both through data/code availability and expanding their methods section.

Below are some remaining concerns I have with the manuscript:

[1] The authors need to explicitly outline the limitations of their analysis with regard to the low power and largely overlapping confidence intervals. The authors point to the preprint by Whiffin et al. using a similar approach. The authors of this previous study clearly stated where results were non-significant and outlined the limitations regarding power in their discussion. The authors here should do similarly.

Specifically:

(a) The authors should state power and low variant numbers across their analyses as a fourth major limitation in their discussion section.

We thank the reviewer for their thoughtful feedback on our manuscript. We agree with the reviewer that a frank discussion of the power limitations used in our MAPS analysis is warranted and we have expanded our discussion to address this point specifically.

(b) The authors spend considerable effort discussing the increased selection against the central G of the triplet compared to the outer Gs. The fact remains that this is not a result shown in the data, as the confidence intervals for all of these analyses are largely overlapping. For the authors to retain this section in the manuscript they should explicitly state that there is no significant difference. They should also be careful with their interpretation of this 'result' in the discussion.

The reviewer makes a valid point here. The confidence intervals for non-central G-tract variants were particularly large because we had removed CpG sites, which disproportionately affected these 5' and 3' flanking variants. With the recent public release of the methylation-adjusted mutation rate tables by gnomAD, we have updated our MAPS model following the procedure outlined in the gnomAD preprint¹ to account for differential methylation levels at CpG sites specifically. Using this updated model, we have comprehensively reanalyzed our original data. Our updated results show that central position guanines across all pG4 contexts are significantly more enriched for rare variation by MAPS compared to the 5' and 3' guanines respectively (permuted P-value = 0.0237, 0.0022 respectively). We have updated the main text, and Figure 1d, Suppl. Figure 4 and our methods sections to include these new results and briefly describe the incorporation of methylation levels into estimating the proportion of singletons expected in the MAPS model.

In addition, the authors should consider referencing this previous work to highlight that a similar approach has been used previously.

We have added a reference to the Whiffin et al. preprint in our results.

[2] I would like to thank the authors for switching from obs/exp to using the LOEUF metric. The authors should, however, include the name 'LOEUF' in the section in brackets to aid readers in comparing across different studies.

We have updated the text to explicitly state that LOEUF is being used.

[3] I am still unclear as to what Figure 1b is showing. Please can the authors explicitly write in the legend what is being plotted on the y axis and make sure this is clear in the main text.

We have updated the legend and the main text to make clear we are plotting the average allele frequency for variants across pG4 vs. non-pG4 forming regions.

[4] The authors talk about assessing "the extent of polymorphism within UTR pG4 sequences". I am not sure that the word 'polymorphism' is the right one here.

We thank the reviewer for pointing this out and have updated the language here.

[5] I think Figure 1c would benefit from lines showing the constraint against synonymous, missense and pLoF variants as well as the existing intergenic line to allow the reader to interpret the level of constraint again pG4s.

We agree with the reviewer that performing this analysis would help place into context the selection acting against UTR pG4 sequences, however we would like to point out that the specific heptamer model we have applied was developed to model mutation rates in noncoding regions. The original publication² includes a separate mutational model that explicitly accounts for average selective pressures within coding regions of the genome (including different amino acid substitutions), however we do not believe that a comparison of observed / expected ratios between these two models is easily interpretable, and performing such a comparison may raise further questions that are beyond the scope of our current study.

[6] Sup. Fig. 4 needs to be referenced in the main text.

We have amended the main results section to include reference to Suppl. Fig. 4.

[7] I appreciate the authors efforts to demonstrate that removing linked SNPs does not remove their eQTL association results. This analysis should be included in the paper – either in the methods or supplement for other readers with similar concerns (apologies if I missed where this has been included).

We have included this analysis in Supplemental Table 3 and amended the main text to reference these additional statistics.

[8] Sup. Table 3 should include numbers for non-pG4 regions as well as pG4s.

We have updated Supplemental Table 3 with numbers for non-pG4 regions.

[9] In the ClinVar enrichment section the sentence “Instead, we hypothesized that variation, in general, is enriched within the UTR pG4 sequences of disease-associated genes” does not appear to make sense. Surely we would expect variation ‘in general’ to be depleted but ‘disease-associated’ variation to be enriched?

We thank the reviewer for highlighting this point of confusion in our manuscript and have rephrased this sentence to clarify our hypothesis in the text.

REFERENCES

1. Karczewski, K. J. *et al.* Variation across 141,456 human exomes and genomes reveals the spectrum of loss-of-function intolerance across human protein-coding genes. *bioRxiv* 531210 (2019). doi:10.1101/531210
2. Aggarwala, V. & Voight, B. F. An expanded sequence context model broadly explains variability in polymorphism levels across the human genome. *Nat. Genet.* **48**, 349–355 (2016).

REVIEWERS' COMMENTS:

Reviewer #2 (Remarks to the Author):

Thank you very much for your detailed response to my queries. I have no further comments.